# Dendritic cell-targeted therapy expands CD8 T cell responses to bona-fide neoantigens in lung tumors

Lucía López [1], Luciano Gastón Morosi [1], Federica La Terza [2], Pierre Bourdely[3,13], Giuseppe Rospo [4,14], Roberto Amadio[1], Giulia Maria Piperno[1], Valentina Russo[5,6], Camilla Volponi[1,15,16], Simone Vodret[1], Sonal Joshi[1], Francesca Giannese [7,8], Dejan Lazarevic[7,8], Giovanni Germano[4,9], Patrizia Stoitzner[10], Alberto Bardelli [4,9], Marc Dalod [11], Luigia Pace[5,6], Nicoletta Caronni [2], Pierre Guermonprez [12] & Federica Benvenuti [1] ✉

Cross-presentation by type 1 DCs (cDC1) is critical to induce and sustain antitumoral CD8 T cell responses to model antigens, in various tumor settings. However, the impact of cross-presenting cDC1 and the potential of DC-based therapies in tumors carrying varied levels of bona-fide neoantigens (neoAgs) remain unclear. Here we develop a hypermutated model of non-small cell lung cancer in female mice, encoding genuine MHC-I neoepitopes to study neoAgs-specific CD8 T cell responses in spontaneous settings and upon Flt3L + αCD40 (DC-therapy). We find that cDC1 are required to generate broad CD8 responses against a range of diverse neoAgs. DC-therapy promotes immunogenicity of weaker neoAgs and strongly inhibits the growth of high tumor-mutational burden (TMB) tumors. In contrast, low TMB tumors respond poorly to DC-therapy, generating mild CD8 T cell responses that are not sufficient to block progression. scRNA transcriptional analysis, immune profiling and functional assays unveil the changes induced by DC-therapy in lung tissues, which comprise accumulation of cDC1 with increased immunostimulatory properties and less exhausted effector CD8 T cells. We conclude that boosting cDC1 activity is critical to broaden the diversity of anti-tumoral CD8 T cell responses and to leverage neoAgs content for therapeutic advantage.

NSCLC tumors carrying elevated tumor mutational burden (TMB) generate neoAgs and correlate to better responses to immune checkpoint inhibitors (ICB)[1–5]. However, actual responses remain suboptimal, even in TMB-high tumors, highlighting the need to unlock neoAgs immunogenicity. cDC1 are critical to cross-present tumor antigens and to induce and sustain anti-tumoral CD8[+] T cell responses, as largely demonstrated by models based on overexpression of artificial antigens[6–12]. In human cancers, cDC1 signatures are associated to improved CD8 T cell responses and better clinical outcomes[13,14]. In addition to cross-presentation, cDC1 contribute to anti-tumoral

immunity by cross-dressing and cytokine-mediated programming[15,16]. However, multiple detrimental axes during lung tumor progression conspire to blunt DC1-mediated antigen presentation[17–21]. A promising way to counteract suppression is to administer Flt3L (Fms-Like tyrosine kinase 3), a growth factor that promotes development and maintenance of the DCs compartment, in combination with adjuvants, radiotherapy, chemotherapy and oncolytic virotherapy, as demonstrated in various preclinical models and ongoing clinical trials[7,9,22–31]. In the context of lung cancer, the NSCLC experimental model KP (Kras[G12D/WT]; Tp53), has been largely shown to be refractory to

---

checkpoint inhibitors[19,21,32,33]. Initial evidence suggests the beneficial roles of Flt3L and αCD40 to revert CD8 T cell suppression in the context of strong model antigens[34].

Here we aim to explore the potential of cDC1 and DC-therapy in a setting that models diverse ranges of bona-fide neoAgs, like those naturally occurring in tumors with different TMB levels. To this goal, we generate a variant of the KP model[35], carrying enhanced mutational load and we identify neoAgs encoded by basal and hypermutated cells. We compare tumor growth, neoAgs-specific CD8 T cell responses and therapy efficacy in TMB low and high tumors, in immunocompetent and cDC1 deficient hosts. We find that cDC1 are required to enhance responses to neoAgs and to promote the immunogenicity of sub-optimal epitopes. Boosting DCs promotes robust anti-cancer responses in highly immunogenic tumors and mobilizes partial responses in less immunogenic tumors.

## Results

### Generation and validation of an NSCLC model carrying increased non-synonymous mutations
To study responses to bona-fide neoAgs in lung tumors, we started by generating a variant of the poorly immunogenic KP model of NSCLC (Kras[G12D/WT]; Tp53) with increased TMB. This was achieved by genetic deletion of *Mlh1*, a critical regulator of mismatch repair, previously shown to increase the rate non-synonymous mutations[32,36] (Fig. 1A, Supplementary Fig. 1A). Two selected Mlh1-deficient KP clones (KP[KO1] and KP[KO2]) were expanded in vitro for several generations and sequenced to compare their mutational profile to that of parental KP cells (KP[ctrl]). Whole exome sequencing showed an increment in single nucleotide variants (SNVs) and frameshifts in both Mlh1 deleted clones (Supplementary Fig. 1B). KP[ctrl] and KP[KO] had similar rates of in vitro proliferation (Supplementary Fig. 1C). However, KP[KO1] lost MHC class-I expression, precluding further analysis (Supplementary Fig. 1D). We thus retained KP[KO2] and named it KP[neo] hereafter. Importantly, gene pathways related to cell proliferation, metabolism, inflammatory responses and antigen processing were not significantly affected in KP[neo], ruling out confounding effects for the following in vivo analysis (Supplementary Fig. 1E). We next implanted KP[ctrl] and KP[neo] cells subcutaneously (s.c.) to assess tumor growth in vivo. Both genotypes grew progressively, however, KP[neo] tumors remained significantly smaller than KP[ctrl]. Depletion of CD8 T cells accelerated the growth of KP[neo], while having a minimal effect on KP[ctrl]. Moreover, KP[neo] tumors implanted in Batf3[−/−] hosts (cDC1 deficient), grew significantly faster than in wild-type, indicating that a cDC1-CD8 axis contributes to anti-tumoral immunity (Fig. 1B, C). Profiling of tumors by flow cytometry confirmed a significant enhancement of total infiltrating CD45[+] cells and increased numbers of CD8[+] T cells in KP[neo] tumors as compared to KP[ctrl], which was abrogated in Batf3 deficient animals (Supplementary Fig. 1F, G). Tissue labeling of tumor sections confirmed that CD8 T cells deeply infiltrate the tumor mass (Supplementary Fig. 1H). Moreover, we detected significantly higher numbers of cDC1 recruited in KP[neo] tumor tissues (Supplementary Fig. 1I). Functionally, CD8 T cells in KP[neo] tumors expressed higher markers of effector memory and PD-1 than in KP[ctrl] (Supplementary Fig. 1J). Consistently, ex-vivo restimulation of CD8 cells from lymph nodes and RNA profiling of tumor tissues showed increased expression of IFN-γ and cytotoxic molecules in KP[neo] (Supplementary Fig. 1K, L). We conclude that inducing non-synonymous mutations in basal KP cells triggers CD8 T cell activation and partial control of tumor growth, mediated by CD8 and cDC1.

### The pattern of tumour-induced T cell responses to neoAgs depend on cDC1
We next investigated the identity and immunogenicity of MHC-I class-I peptides expressed basally in KP[ctrl] and those generated de novo in KP[neo]. Using the well-established predictor netMHCpan 4.0 on whole exome data, followed by RNAseq validation, we identified 22 neoAgs expressed in both KP[ctrl] and KP[neo] cells (shared neoAgs) plus 26 additional neoAgs acquired by KP[neo] (unique neoAgs), featuring various affinities and expression levels (Fig. 1D, Supplementary Table 1). To examine the specificity of T cell responses for the identified neoAgs, CD8 T cells from the spleen of mice injected with KP[ctrl] or KP[neo] tumors were restimulated ex-vivo with selected peptides and interferon-γ (IFN-γ) secretion was measured by enzyme-linked immunospots (ELISpot). KP[neo] tumors induced clear responses to several unique peptides (Fig. 1E). As expected, KP[ctrl] tumors induced no response to unique neoAgs, validating the specificity of detection. Moreover, KP[ctrl] did not respond to shared peptide, confirming low immunogenicity (Fig. 1F). Interestingly, however, we detected clear responses to shared neoAgs in hosts challenged with KP[neo] tumors, demonstrating epitope spreading and induction of reactivity to otherwise immunologically cold neoAgs (Fig. 1F). To confirm reactivity to shared neoAgs, we functionalized H-2D[b] custom dextramers with shared peptide 1 to evaluate the presence of dextramer-positive CD8[+] T cells in tumor tissues of mice challenged with KP[ctrl] or KP[neo] tumors, respectively. Peptide-specific CD8[+] T cells were detected in KP[neo] tumors but not in KP[ctrl], in line with ELISpot results (Fig. 1G). To investigate the role of cDC1 in determining epitope spreading, we performed the same analysis in Batf3 deficient mice. Remarkably, the magnitude of responses to both primary immunogenic antigens and secondary antigens rescued by spreading was blunted in the absence of cDC1 (Fig. 1H). We conclude that cDC1 are critical to enhance and broaden responses to less immunogenic antigens.

### DC-therapy amplifies responses to weak neoAgs and eradicates TMB[high] tumors
KP tumors are refractory to various combinations of checkpoint inhibitors, even in the presence of strong immunogenic antigens, highlighting the need to implement alternative or additional axes[19,21,32]. In agreement with these previous reports, αPD-L1 treatment had no impact on the growth of KP[ctrl] or KP[neo] tumors (Fig. 2A). Based on the findings presented in Fig. 1, we reasoned that boosting DCs may further diversify and enhance responses to less immunogenic epitopes for clinical benefit. We thus challenged mice with KP[ctrl] or KP[neo] tumors and delivered a combination of Flt3L to expand DCs plus αCD40 to activate them (FL/αCD40, DC-therapy)[7,34]. Interestingly, DC-therapy, in the absence of any T cell targeting molecule, induced rejection of the majority of KP[neo] tumors (Fig. 2B, C). Conversely, poorly immunogenic KP[ctrl] tumors grew progressively under DC-therapy, albeit more slowly than the untreated group (Fig. 2B, C). To investigate immune changes underlying these results, we first verified expansion and activation of cDC1 at day 12 after tumor implantation, when therapy was discontinued. As tumors were too small to reliably detect cDC1, we quantified expansion in lymph nodes. DC-therapy induced significant cDC1 expansion and upregulation of maturation markers in both tumor genotypes (Supplementary Fig. 2A, B). At the same time point (day 12), when tumor lesions were comparable in all groups, we observed upregulation of effector T cell markers in both the tumor-draining and contralateral lymph nodes, similarly in KP[ctrl] and KP[neo], indicating that initial activation occurs in the two genotypes (Fig. 2D, Supplementary Fig. 2C). Conversely, frequencies and absolute numbers of CD8 T cells in tumor tissues were significantly larger in KP[neo] and further expanded by therapy. CD8 T cells infiltrating KP[ctrl] were also modestly increased by therapy, yet their numbers and frequencies remain significantly smaller than in KP[neo] (Fig. 2E). Analysis of the immune composition at day 21, confirmed that residual lesions in the KP[neo]-treated group were highly infiltrated by CD8 T cells, whereas grown-up KP[ctrl] tumors contain relatively few CD8 T cells (Fig. 2F). Importantly, the CD8/Treg ratio was significantly increased in KP[neo]-treated tumors at the endpoint as compared to the non-treated group

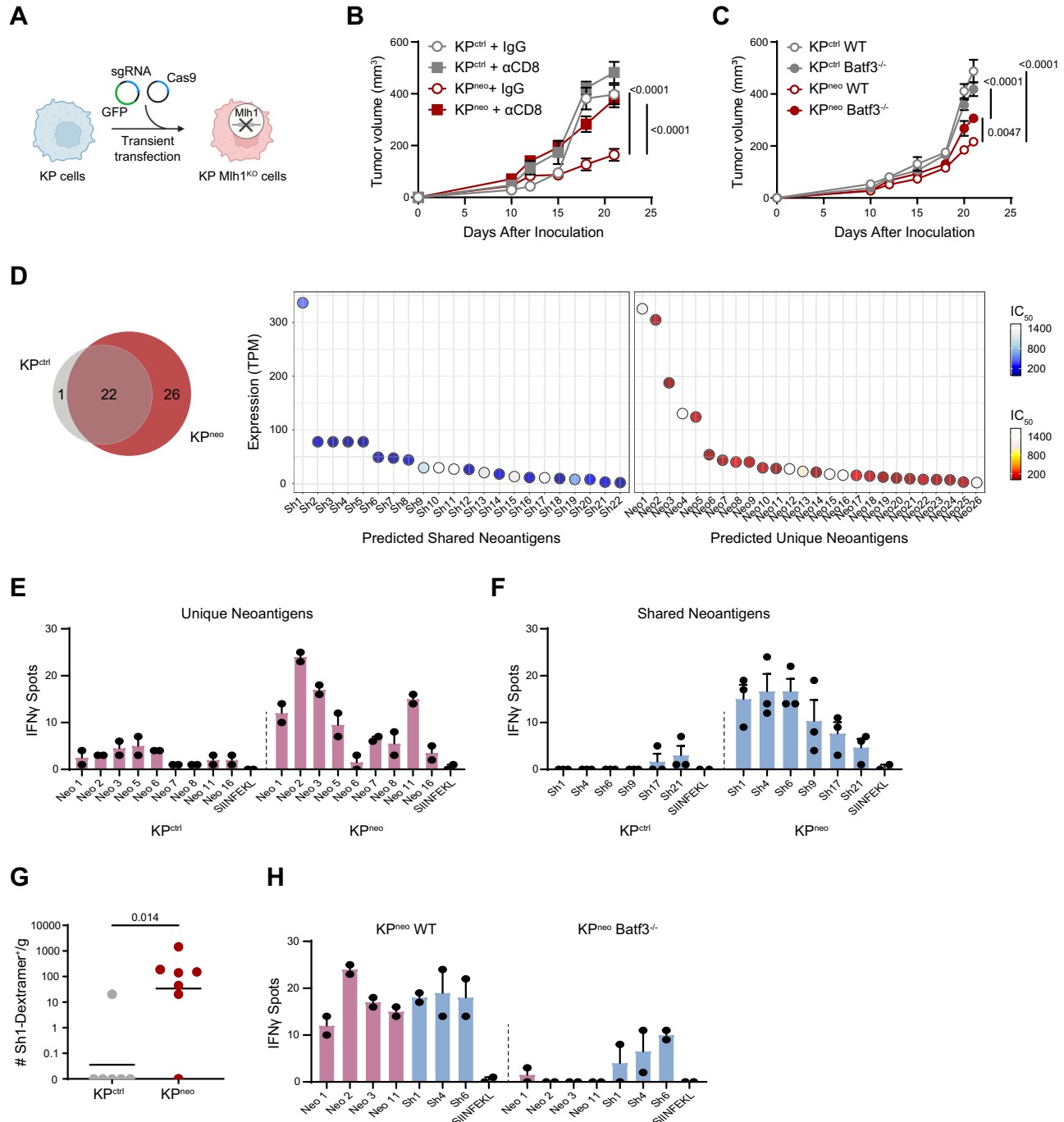

**Fig. 1 | Increasing neoAgs in KP drives cDC1-dependent anti-tumoral responses to neoAgs. A** KP cells were transiently transfected with CAS9 and sgRNA targeting the Mlh1 locus to generate Mlh1-deficient cells, or CAS9 alone (KP^ctrl) (Created with BioRender.com). **B, C** $5 \times 10^5$ KP^ctrl or KP^neo tumor cells were implanted subcutaneously, and tumor outgrowth was measured at the indicated time points. **B** Mice were treated with anti-CD8 antibodies or isotype control (IgG) ($n = 5$, data are from one out of the two independent experiments). **C** Tumor cells were implanted into wild-type or Batf3^−/− ($n = 6$, data are from one out of two independent experiments). **D** The Venn diagram represents the number of shared and unique neoantigens expressed at mRNA level in KP^ctrl and KP^neo cells. neoAgs commonly expressed in KP^ctrl and KP^neo (shared, sh) and neoAgs expressed de-novo in KP^neo (unique, neo) are plotted according to expression levels (TPM) and affinity for MHC class-I (IC_{50}). **E, F** At day 12 after challenge, splenic CD8 T from KP^ctrl or

KP^neo tumor-bearing mice were restimulated with selected unique (**E**) or shared (**F**) individual neoAgs and tested for IFN-g by ELISpot. Individual dots are technical replicates from one representative experiment (pooled CD8 T cells from 3 mice), out of three performed. **G** KP^ctrl or KP^neo tumor tissues were harvested at day 26 and labeled with sh1 MHC class I-specific dextramers to identify neoAg-specific CD8^+ T cells. Data from 1 experiment with 6 animals for KP^ctrl and 7 animals for KP^neo group. **H**) Wild type or Batf3^−/− mice were challenged with KP^neo tumors. At day 12, CD8 T cells were isolated from the spleen to test reactivity against selected shared and unique peptides by IFN-γ ELISpot. Individual dots are technical replicates from one representative experiment (pooled CD8 T cells from 3 mice), out of three performed. Two-way ANOVA followed by Sidak's post-test in (**B**), or Tukey's post-test in C, two-tailed Mann-Whitney $U$ test in (**G**). All data are plotted as mean ± SEM. Source data are provided as a Source Data File.

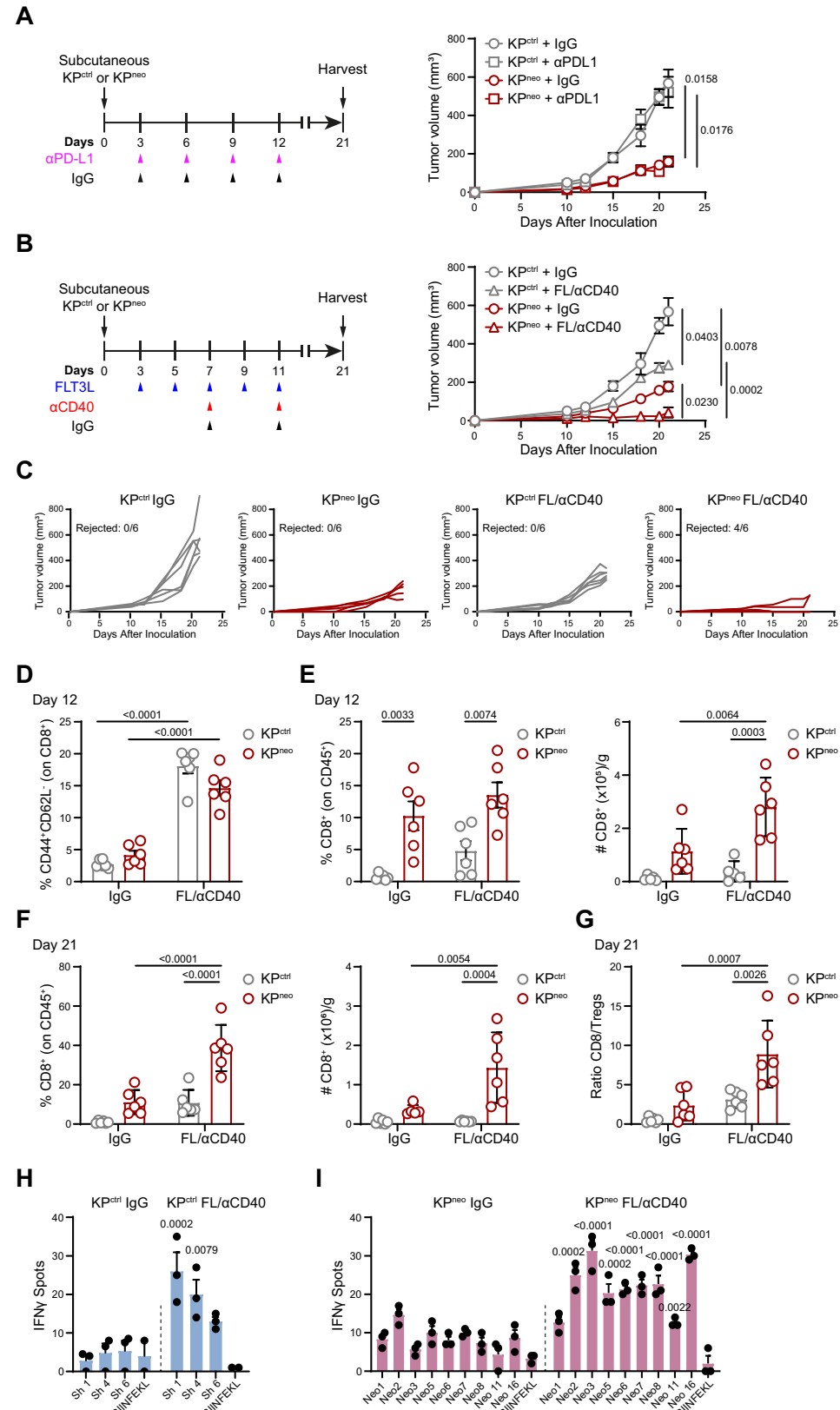

and KP^ctrl (Fig. 2G). The CD4 compartment was more abundant in KP^neo than in KP^ctrl, both at initial stages and at the endpoint, but it was not expanded by therapy (Supplementary Fig. 2D, E). To understand the specificity of antitumoral responses triggered by DC-therapy, we performed ELISpot analysis on a selection of shared and unique peptides. Shared peptides lacking basal reactivity in KP^ctrl hosts induced

responses post DC-therapy, consistent with the partial therapeutic effect (Fig. 2H). Of note, therapy triggered robust de novo reactivity to several KP^neo unique peptides that were not basally immunogenic (Fig. 2I). We conclude that DC therapy can unlock the immunogenicity of weak antigens in low antigenic tumors, inducing a partial control of otherwise cold tumors. Remarkably, when neoepitope density is

**Fig. 2 | Flt3L/αCD40 therapy induces regression of KP^neo tumors and has a mild impact on KP^ctrl tumors. A** Scheme of anti-PD-L1 (αPD-L1) or IgG treatment in KP^ctrl or KP^neo tumors (left). Tumor outgrowth (right). **B** Flt3L and anti-CD40 (FL/αCD40) therapy or IgG administration scheme in KP^ctrl and KP^neo tumors (left). Tumor outgrowth (right). **C** Growth of individual tumors showed in (**B**), indicating the fraction of rejected tumors in each group. **D** Frequencies of effector/effector memory (CD44^+/CD62L^-) CD8 T cells in the tdLN at day 12 after tumor challenge.
**E** Frequencies (left) and absolute numbers (right) of tumor-infiltrating CD8 T cells at day 12. **F** Percentage (left) and absolute numbers (right) of tumor-infiltrating CD8 T cells at the endpoint (day 21). **G** Tumor CD8/Treg ratio at day 21. **A–G** Data from one out of two independent experiments is presented (*n* = 6 mice per group, per experiment). **H, I** IFN-γ ELISpot showing the specificities of CD8 T cells under FL/αCD40 therapy, induced by KP^ctrl tumors and tested for shared peptides (**H**) or by KP^neo tumors and tested for unique neoAgs (**I**). *n* = 3, data represent spots from pooled CD8 T cells from 3 mice per group, one out of three independent experiments. Two-way ANOVA followed by Tukey's post-test in A-G; two-tailed Multiple *t-test* with Holm-Sidak correction method in (**H, I**). All data are plotted as mean ± SEM. Source data are provided as a Source Data File.

higher, boosting immunogenicity by DC-therapy is sufficient to induce tumor rejection.

## cDC1 are required for anti-tumoral responses induced by Flt3L/αCD40 therapy

We next explored whether the efficacy of FL/aαCD40 therapy depends on Batf3 cDC1. Delayed progression of KP^ctrl tumors by therapy was intially lost in Batf3^-/- hosts, however, we observed growth arrest at the endpoint, indicating that tumor control is not entirely cDC1 dependent in this context (Fig. 3A). More compelling, the regression of KP^neo induced by FL/αCD40 was completely abrogated in cDC1 deficient animals (Fig. 3B). Accordingly, the recruitment of effector CD8 T cells in KP^neo tumors was abolished in Batf3^-/- animals (Fig. 3C, D). Expression of cytotoxic genes in tumor tissues, which was enhanced by therapy in KP^neo tumors, was abrogated in Batf3^-/- recipient (Fig. 3E). We then examined whether cDC1 regulates therapy-driven responses to neoAgs induced by KP^neo tumors. To this goal, we compared responses to unique neoAgs using CD8 T cells from the spleen of therapy-treated wild type or Batf3^-/- mice implanted with KP^neo. In line with the above data, we detected significantly smaller responses in Batf3 deficient animals, indicating DC-therapy expands neoAgs-specific responses via cross-presentation by cDC1 (Fig. 3F). Altogether we conclude that cDC1 are required for the efficacy of Flt3L-based therapy and contributes to expand the intensity and diversity of CD8^+ T cell responses to neoAgs.

## TMB and cDC1 correlations with CD8 T cells effector scores and prognosis in human NSCLC

To explore the relation between cDC1, neoantigens and anti-tumoral immune responses in human cancers, we stratified lung cancer patients from the Cancer Genome Atlas (TGCA) according to the levels of TMB (Fig. 4A), as a proxy of neoAgs content[37]. We found that a cDC1 signature and a CD8^+ effector T cell signature correlated positively across TMB levels (Fig. 4B). We then split LUAD patients into 4 groups according to low or high cDC1 content (bottom or top quartiles, Fig. 4C) and low or high TMB (below and above median, Fig. 4A). This stratification uncovered that low cDC1 density correlates to poor CD8^+ effector T cells, irrespective of TMB levels, indicating that high TMB is not sufficient to generate effector CD8 T cells when cDC1 are scarce. In contrast, CD8^+ T cell scores were significantly higher in the cDC1 top group and were maximal in the cDC1^high/TMB^high group (Fig. 4D), indicating that high TMB and high cDC1 may synergize to promote CD8 T cell activation. In line with past reports[13,14], cDC1 density was the main factor determining survival showing a slightly more significant p-value in the TMB high group. These data suggest that, across the wide diversity of neoAgs levels in human lung tumors, the impact of cDC1 on CD8 T cell activation is proportional to antigen density. In terms of patient outcome, cDC1 and TMB act as independent variables, indicating that cDC1 can slightly contribute to better responses even in TMB-low tumors (Fig. 4D).

## NeoAgs density determines responsiveness to FL/αCD40 therapy in lung orthotopic tumors

The above observations indicate that KP generates spontaneous and therapy-dependent responses to neoepitopes when tumours are implanted in the mouse flank. A recent report demonstrated that flank

and lung orthotopic tumors generate qualitatively different CD8 T cell responses, with lung environment being more suppressive and refractory to therapy, even in the context of strong model antigens[19]. We first evaluated responses to PD-L1 treatment in KP^ctrl and KP^neo. In line with data in the s.c. setting and previous reports, both models were refractory to checkpoint inhibition (Supplementary Fig. 3A). We next investigate DC-therapy in the context of orthotopic lung tumors to explore if this could override dysfunctional CD8 T cells. Animals carrying KP^ctrl or KP^neo lung tumors were treated with FL/αCD40 or isotype control, according to the scheme depicted in Fig. 5A. Growth of KP^neo in the lungs of untreated mice was slightly delayed as compared to KP^ctrl, indicating that spontaneous immunogenicity is maintained also in lung tissues. Importantly, therapy induced a significant reduction of KP^neo tumors whereas growth of KP^ctrl remained unaffected (Fig. 5B). To verify the effects of systemic FL/αCD40 therapy we examined cDC1 numbers and distribution in lung tissues. Flow cytometry indicated a robust increase in lung resident cDC1 in treated mice, regardless of the tumour genotype, which is consistent with a genotype-independent impact of therapy on cDC1 expansion. Moreover, to precisely visualize cDC1 in lung tumor tissues we implanted KP^neo tumours in XCR1-Venus animals[38]. We detected Venus^+ cDC1 enriched in residual nodules of KP^neo-treated lungs, suggesting that therapy-induced cDC1 infiltration in tumor areas (Fig. 5C, D). Analysis of infiltrating T cells revealed an increment in the fraction and absolute numbers of CD8 T cells upon therapy in lungs carrying KP^neo tumours which was not observed in KP^ctrl (Fig. 5E). We also detected a substantial increment of NK cells infiltrating the lung post-therapy, exclusively in KP^neo (Supplementary Fig. 3B). Conversely, CD4 T cells were not significantly modified (Supplementary Fig. 3C). Immunohistochemistry on tissue sections showed a rich CD8 T cells infiltrate recruited into residual nodules of KP^neo-treated tumors, whereas tumor nodules in KP^ctrl remained mostly immuno-excluded (Fig. 5F, G). To assess CD8 T cell functionality, we analyzed expression of PD-1 and granzyme. Both parameters were slightly, but not significantly, elevated in CD8 T cells infiltrating KP^neo lungs at steady state, indicating antigen exposure and mild spontaneous activation. Therapy enhanced the fraction of PD-1 expressing cells, proportionally in both tumor genotypes and expanded granzyme-producing cells, particularly in KP^neo (Fig. 5H, I). Moreover, the fraction of IFN-γ secreting cells was increased by therapy exclusively in the KP^neo group (Fig. 5J). To assess the specificity of responses we analyzed reactivity of CD8 T cells toward unique neoAgs in KP^neo challenged animals by ELISpot. We detected amplification of specific responses to some but not all neoAgs, with a trend that was similar to that in the s.c. model (Supplementary Fig. 3D). We conclude that FL/αCD40 therapy drives expansion, intratumoral recruitment and functional activation of CD8 cells, to an extent that is proportional to the neoAgs content. These responses can inhibit the progression of more immunogenic lung tumors but are insufficient to block poorly immunogenic tumors in lung tissue.

## DC-therapy expands proliferating effector CD8^+ T cells in lung tissues

To further elucidate molecular changes underlying the beneficial impact of FL/αCD40 therapy in the context of KP^neo tumors, we isolated CD45^+ cells from the lungs of the responder group (KP^neo FL/

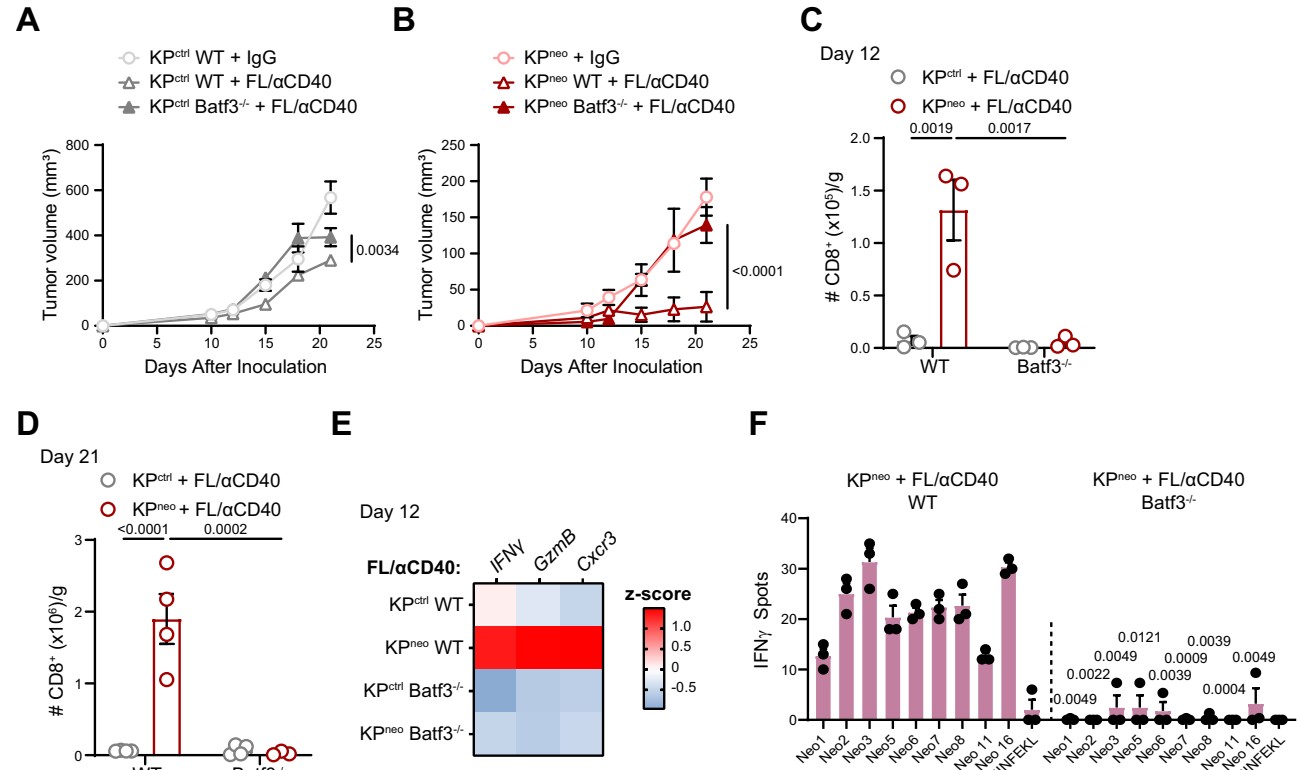

**Fig. 3 | The impact of FL/αCD40 therapy requires Batf3 cDC1. A, B** KP^ctrl (**A**) and KP^neo (**B**) were implanted in wild-type or Batf3^−/− animals and treated with FL/αCD40 as depicted in 2 A. Tumor outgrowth (*n* = 4 mice per group). **C, D** Absolute numbers of tumor-infiltrating CD8 T cells at day 12 (*n* = 3 mice per group) and 21 (*n* = 4 mice per group), respectively. **E** Relative expression of the indicated genes (RT-qPCR) in KP^ctrl and KP^neo tumors tissues upon treatment with FL/αCD40, in WT and Batf3^−/− mice. Heatmap showing the z-score of the indicated genes (*n* = 4). **F** IFN-γ ELISpot showing the specificities of CD8 T cells to unique peptides, induced by KP^neo tumors under DC-therapy in WT and Batf3^−/− mice, respectively. *n* = 3, data represent spots from pooled CD8 T cells from 3 mice per group. Two-way ANOVA followed by Sidak's post-test in (**A**–**C**); or Tukey's post-test in D; two-tailed Multiple *t*-test with Holm-Sidak correction method in F. All data are plotted as mean ± SEM and represent one out of two independent experiments. Source data are provided as a Source Data File.

αCD40) and the control untreated group (KP^neo IgG) to perform scRNA sequencing. We obtained data from 8760 and 6699 total cells in the control and treated conditions, respectively. Unsupervised clustering of integrated data identified 11 clusters that were annotated according to literature-based signatures (Supplementary Fig. 4A, B, Supplementary Data 1). The identified immune populations were consistent with the structure of lung immune populations previously identified in the same tumor model[39] (Supplementary Fig. 4A). Next, we computationally re-clustered T lymphocytes in 10 transcriptionally defined clusters corresponding to CD4 (C0, C2), Tregs (C9), CD8 (C1, 3, 5, 6), γδ-T cells (C7) and MAIT T cells (C8). C4 and C10 were excluded based on the expression of NKT cell markers (*Xcl1, Klra7, Klrb1c, Klra1*) or contaminants (Supplementary Fig. 4C; Supplementary Data 2). By focusing on CD8^+ T cells, we identified four clusters corresponding to different functional states, including naïve T cells (*Sell, Lef1, Ccr9*, C1); transitional naïve to activated/short-lived effector T cells (*Ifit1, Ifit2, Eomes*, C3); conventional activated/exhausted T cells (*Gzmb, Gzmk, Cxcr3, Ccl5, Iga1, Lag3, Pdcd1*, C5) and proliferating/effector T cells (*Top2a, Mki67, Klrg1, Gzma*) (Fig. 6A). Analysis of sample composition revealed a substantial enrichment in C6 and a moderate increase in C5 upon therapy (Fig. 6B, Supplementary Fig. 4D). Differential expressed genes (DEG) between FL/αCD40 and isotype samples uncovered upregulation of cell cycle genes (*Stmn1, Ubec2, Birc5, H2-Q10, Cdca8, Cks1b, Cenpv*) across the 3 non-naïve CD8^+ T cell clusters, particularly in C5 and upregulation of short-lived effectors and cytotoxic genes (*Gzma, Serpina3g, Klrg1, Eomes, Gzmk*) upon therapy. In addition, C3 selectively upregulated genes associated to resident effector memory programs (*Ifitm3 and Lgals3*)[40,41], and C5 selectively upregulated Tpx2, which was recently associated to anti-tumoral CD8^+

T cells[42]. Genes defining resident memory features resulted as negatively modulated in C3 and C5 after therapy (*Lars2, Pmepa1a, Itgb1* and *Icos*), likely reflecting the upregulation of effector functions. Similarly, the highly proliferative subset C6 showed reduced expression of *Ccr4, Cxcr6, Rbpj, Tnfrsf4* upon therapy, reflecting a less differentiated phenotype (Fig. 6C, Supplementary Data 3). GSEA analysis indicated specific enrichment in oxidative phosphorylation, aerobic respiration, mitochondrial function and cell division in C3, C5 and C6 after therapy (Supplementary Fig. 4E, Supplementary Data 3). Interestingly, a gene signature defining T cell activation and cytotoxicity (*Gzma, Gzmb, Prf1, Tnf, Ifng, Cxcr6*)[43], was significantly upregulated after therapy in all three non-naïve CD8^+ T cell clusters, more prominently in C5 and C6. Conversely, genes associated with T cell exhaustion were significantly downregulated upon therapy in C5 and C6 (Fig. 6D). The two CD4 clusters, C0 and C4 showed upregulation of cytotoxic genes and modulation of metabolic processes upon therapy but were not further analyzed here (Supplementary Data 4).

To validate scRNA findings in CD8 T cells, we next examined the most relevant changes by flow-cytometry in KP^neo and included KP^ctrl, for comparison. The frequency of proliferating CD8 T cells (Ki67^+) was increased almost equally in KP^ctrl and KP^neo, however, absolute numbers of proliferating CD8 T cells remained consistently higher in KP^neo (Fig. 6E). Moreover, effector functions (IFN-γ production) and the frequency of stem-like TCF-1^+PD1^+CD8^+ T cells[44], were preferentially induced in KP^neo post-therapy (Fig. 6F, G). Importantly, Tim3^+Lag3^+ cells reflecting chronic antigen exposure and exhaustion were higher in KP^neo, but diminished upon therapy (Fig. 6H). By intravenous delivery of a pan-leukocytes antibody we excluded circulating cells and confirmed increased proliferation, granzyme

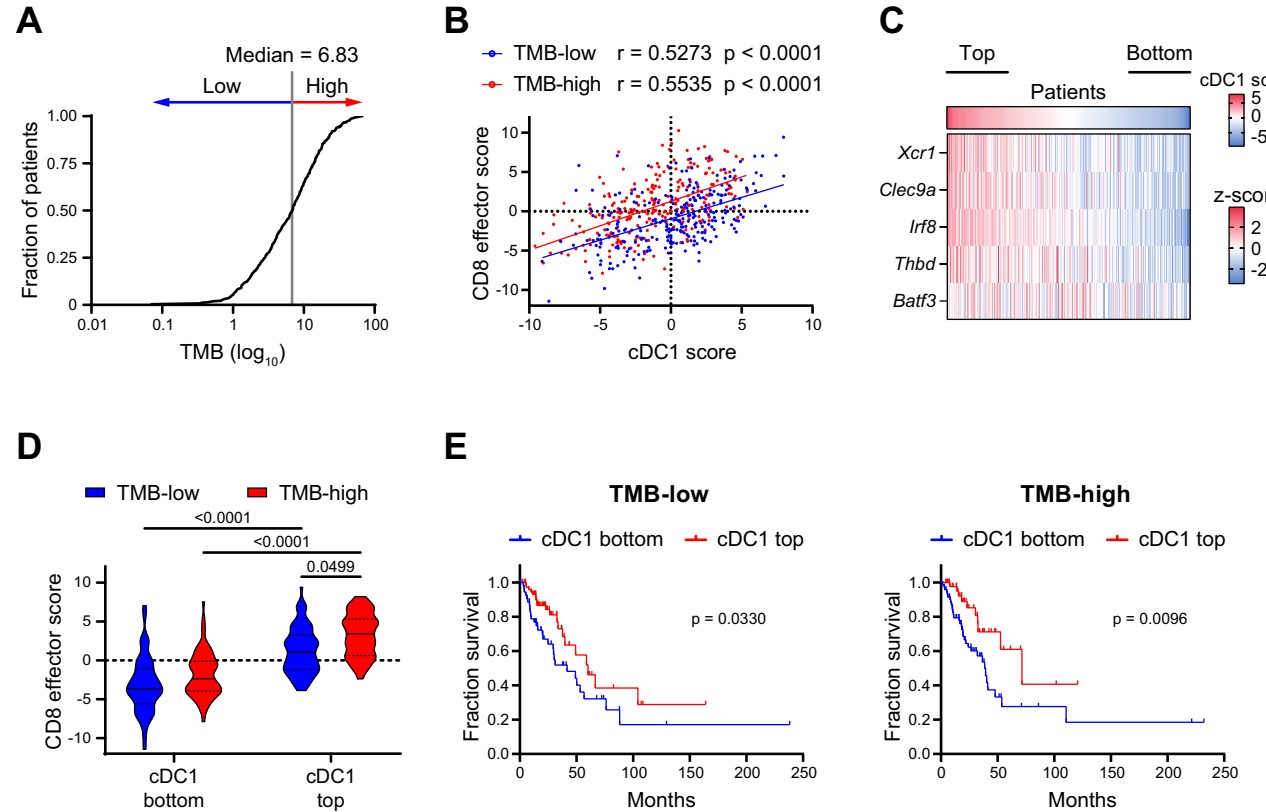

**Fig. 4 | cDC1 positively correlates with effector CD8 T cells in LUAD across a range of TMB values. A** TMB distribution across lung cancer patients in the TCGA-LUAD PanCancer Atlas dataset. The median value was used to stratify patients as TMB-low or TMB-high. **B** Correlation between cDC1 and CD8 effector T cell scores. Correlation analysis was performed by two-tailed Pearson correlation coefficient (r). **A, B** Total number of patients=506; where, TMB-high=253 and TMB-low=252. **C** Heatmap showing the expression of the cDC1 signature genes across the dataset (z- score average for the indicated genes). cDC1 bottom and top quartiles are highlighted (*n* = 127 each). **D** Violin plots showing CD8 effector T cell scores for bottom and top cDC1 quartiles within TMB-low and TMB-high groups. Data represent median (continuous line) and interquartile (dashed lines) (For TMB-low and cDC1 bottom group *n* = 54, TMB-low and cDC1 top *n* = 79, TMB-high and cDC1 bottom *n* = 73, and TMB-high and cDC1 top *n* = 48). Statistical analysis was performed by two-way ANOVA followed by Tukey's posttest. **E** Prognostic values of the cDC1 signature for cancer patient overall survival (on months) comparing bottom and top cDC1 quartiles in the TMB-low and TMB-high groups (TMB-low, *n* = 127; TMB-high, *n* = 127). Survival curves were compared using the Log-rank test (Mantel-Cox). Source data are provided as a Source Data File.

production and loss of exhaustion in the extravascular lung CD8 T cell fraction (Supplementary Fig. 5A–D). Altogether these data indicate that DC-therapy promotes the *novo* differentiation and expansion of neoAgs specific, not exhausted, effector cells endowed with cytotoxic functions.

### FL/αCD40 therapy induces remodeling of the cDCs compartment in lung tissues

Next, we focused on the DCs compartment at a higher resolution to examine functional changes induced by therapy. Re-clustering of cDCs identified 4 main DC subsets (C0-C3, Supplementary Fig. 6A): C0 express DC2 markers (*Itgam, CD209* and *Sirpa*); C1 and C2 express cDC1 markers (*Xcr1, Tlr3, Clec9*), C3 formed a well-defined separated cluster marked by prominent expression of *Ccr7, Fscn1* and *Socs2*, coinciding with the subset of mature DCs enriched in regulatory markers (mregDCs)[21] (Fig. 7A, B; Supplementary Data 5). Cells in C1 (named cDC1a hereafter) were classified as differentiated cDC1 based on high expression of cDC1 markers and biological processes associated to mature cDC1 functions; cells within C2 showed lower expression of cDC1 makers and functional processes but were enriched in cell cycle genes (*Mik67* and *Topa2*) and were classified as pre-cDC1 progenitors and named cDC1b[45] (Fig. 7A, B, Supplementary Fig. 6B, Supplementary Data 5). Analysis of sample and cluster composition uncovered that FL/αCD40-treated samples were enriched in cDC1 clusters (C1 and C2) and depleted in mregDCs (Supplementary Fig. 6C, D). Flow cytometry confirmed a significant reduction of

CD11c+/MHC-II+/CCR7+/PD-L1+ DCs after therapy. Loss of the population was occurring as well in the context of KP[ctrl], suggesting it is intrinsic to DC remodeling and not dependent on neoAgs density (Supplementary Fig. 6E). DEG analysis identified genes modulated by therapy in cDC1a, cDC1b and cDC2, whereas mRegDCs were too scarce after therapy to allow comparison (Supplementary Fig. 6F, Supplementary Data 6). DEGs analysis revealed that some regulatory genes defining the mreg program (*Axl, Aldh1, Cd274*)[21], were downregulated across the 3 subsets (Supplementary Data 6). We compiled a signature that includes regulatory and inhibitory DCs genes (*Axl, Ccl22, Ccl17, Cd274, Aldh1a1, IL10ra, Tgfbr1*) to define DCs exhaustion. Interestingly, the exhausted DCs program was highly expressed in untreated lung tumors and significantly diminished in FL/αCD40 treated samples (Fig. 7C). GSEA highlighted enrichment in genes controlling responses to IFN-β selectively in cDC1a, which may reflect the specialization of lung cDC1 in type-I IFN sensing[46]. Moreover, enrichment of pathways related to chromatin assembly, RNA processing, mitochondrial function and cholesterol biosynthesis were enriched in the three subsets, more prominently in cDC1a, consistent with the higher mitochondrial mass and specific lipid metabolism[47,48] (Supplementary Fig. 6G, Supplementary Data 6). To verify whether transcriptional changes emerging by scRNAseq correspond to cellular and functional remodelling of cross-presenting cDC1, we implanted KP[neo] tumors in XCR1-Venus mice. Venus fluorescence was restricted to a defined population of CD11c+MHC-II+XCR1+ lung cells, unequivocally defining cDC1 (Supplementary Fig. 6H). Therapy induced significant expansion of a

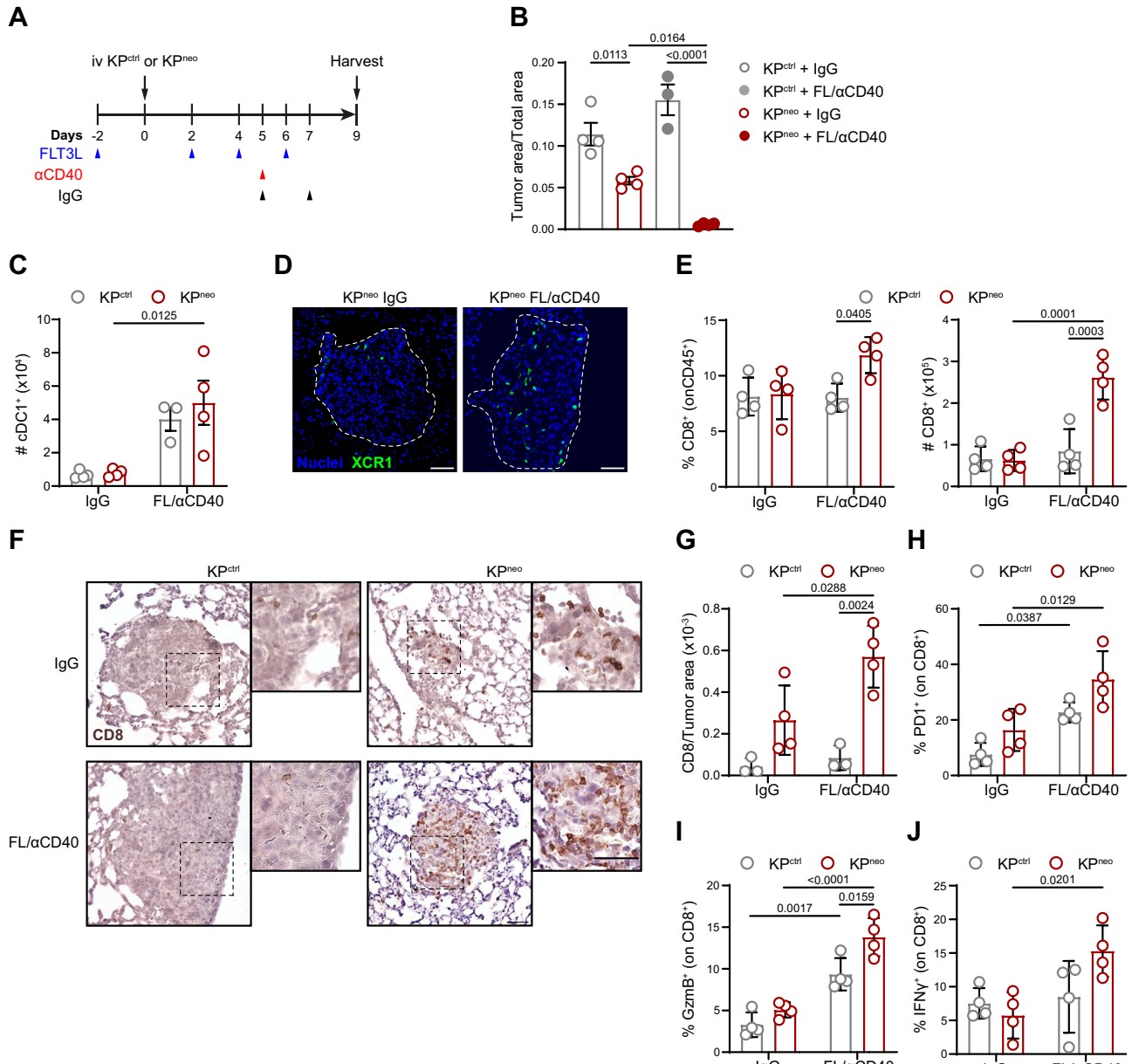

**Fig. 5 | FL/αCD40 therapy triggers CD8 T cell responses and tumor regression in KPneo lung orthotopic tumors. A** KP[ctrl] and KP[neo] tumors were implanted orthotopically in the lung and treated as depicted in the scheme. **B** Quantification of tumor burden 9 days after tumor challenge. Tumor burden was calculated as the ratio between tumor nodules and total lung area ($n = 4$). **C** Absolute numbers of cDC1 in tumor-bearing lungs with KP[ctrl] or KP[neo] tumors upon FL/αCD40 or IgG quantified by FC ($n = 4$). **D** Representative tissue cryosections showing localization of cDC1 within KP[neo] tumor nodules (dotted lines) in XCR1-Venus mice. Representative images from one out of four animals per group, from one independent experiment out of three performed. Scale bars represent 50 μm. **E** Frequencies (left) and absolute numbers (right) of CD8 T cells in lung tissues were quantified by FC ($n = 4$). **F** Representative IHC images of lung tissues labeled with anti-CD8

antibodies, from one out of four animals per group, from one independent experiment out of three performed. Insets depict CD8 T cells infiltrating tumor nodules. Scale bars represent 50 μm. **G** Quantification of the number of CD8[+] cells infiltrating lung nodules showed in (**F**) ($n = 4$). **H, I** Relative frequencies of PD1[+] CD8[+] T cells (**H**) and granzyme B (GzmB)-producing CD8 T cells (**I**) in lungs of animals carrying KP[ctrl] or KP[neo] tumors, treated with therapy or IgG ($n = 4$). **J** Relative frequencies of IFN-γ-producing CD8 T cells in the lungs of the indicated groups were measured after ex-vivo restimulation and ICS ($n = 4$). Two-way ANOVA followed by Tukey's post-test in (**B, E, G, H, I, J**); or Sidak's post-test in (**C**). All data are plotted as mean ± SEM, and represent one out of two independent experiments. Source data are provided as a Source Data File.

population expressing intermediate levels of Venus fluorescence (Venus[int]), distinguished from Venus[high] cells (Fig. 7D), reflecting the transcriptional profile of cDC1b and cDC1a, respectively. Moreover, Ki67 labeling showed that Venus[int] cells were extensively proliferating post-therapy, recapitulating the phenotype of committed pre-DC precursors[45] (Fig. 7E). In terms of functionality, we detected a small fraction of IL-12 producing cells within the Venus[high] population after therapy, in agreement with their identity as mature cells and the action

of αCD40L (Fig. 7F). Conversely, expression of the inhibitory receptor PD-L1, which was detected on both Venus[high] and Venus[int] cells, was significantly reduced by therapy (Fig. 7G). Tim3, an inhibitory checkpoint on cDC1[49], was expressed on Venus[high] cells and diminished by therapy (Fig. 7H). Finally, we cell sorted Venus[+] cells from the lungs of therapy- or isotype-treated KP[neo] tumors to directly assess their immunostimulatory properties. To facilitate detection, we pulsed DCs with increasing concentrations of OVA MHC-I peptide and mixed them

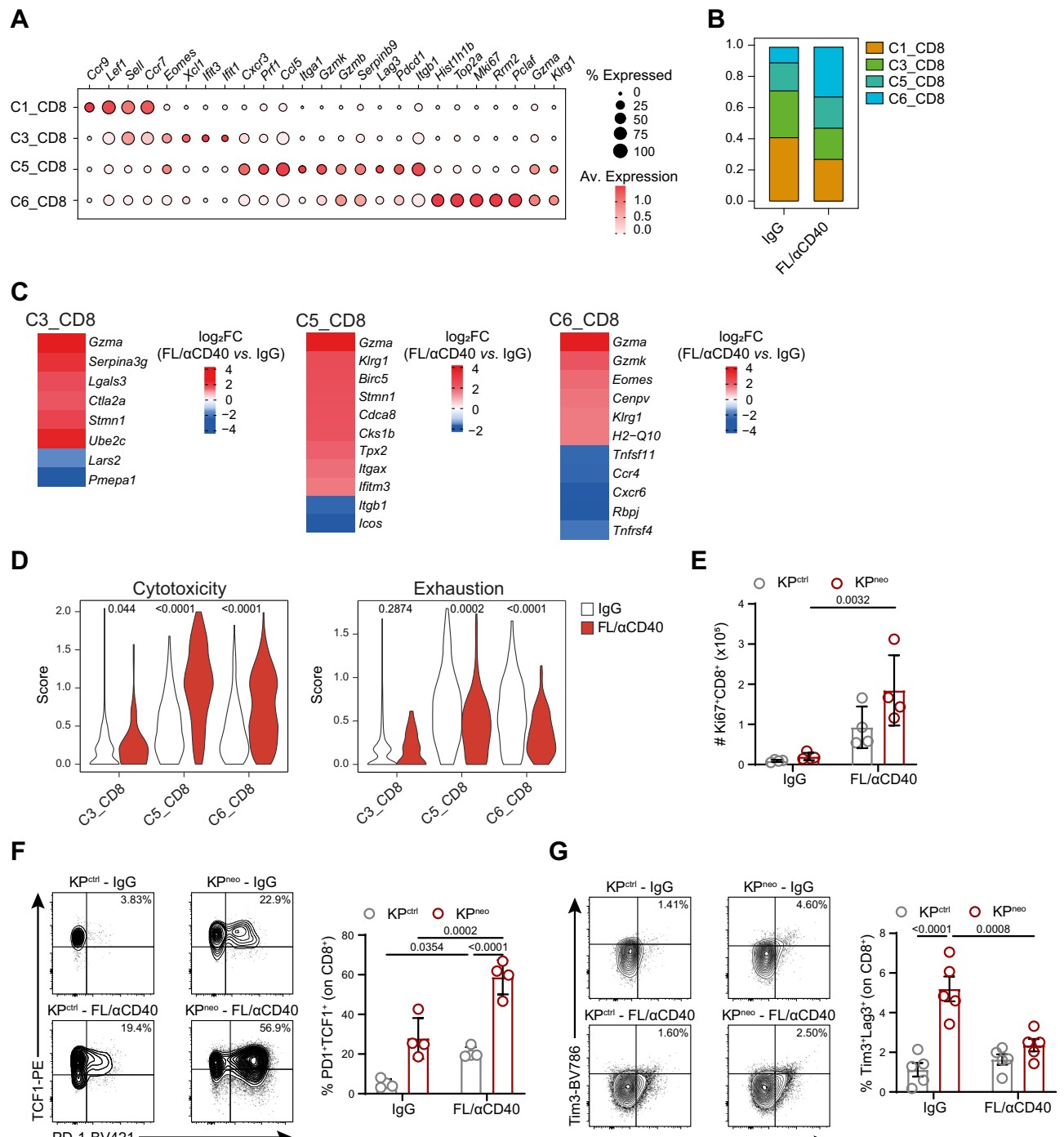

**Fig. 6 | DC therapy relieves exhaustion of CD8⁺ T cells in lung orthotopic tumors.** Mice were implanted with orthotopic KPⁿᵉᵒ tumors and treated with FL/αCD40 or IgG, CD45⁺ cells were isolated from the lungs at day 9 post-tumor challenge and subjected to scRNA-seq. **A** Dot plot showing scaled gene expression of selected genes defining CD8⁺ T clusters. **B** Proportion of CD8 T-cell clusters. **C** Heatmaps showing log₂FC of selected differentially expressed genes (DEG) in FL/αCD40 *vs.* IgG in C3, C5 and C6 clusters (average log₂FC FL/αCD40 vs. IgG FDR < 0.01). **D** Violin plots show combined mean expression values for the indicated genes (score) for the cytotoxic T cell markers (*Gzma, Gzmb, Prf1, Tnf, Ifnγ, Cxcr6*) or

exhaustion markers (*Pdcd1, Ctla4, Tigit, Lag3, Havcr2, Tox, Nrp1*) in cells of the indicated clusters in FL/αCD40 *vs.* IgG. **A–D** scRNAseq data correspond to a pool of 4 animals per group. **E** Absolute numbers of Ki67⁺ CD8 T cells evaluated by ICS on KPᶜᵗʳˡ- and KPⁿᵉᵒ-bearing lungs (*n* = 4). **F**, **G** Representative FC plots and quantification of the frequencies of TCF-1⁺PD-1⁺ (**F**, *n* = 4 mice) and Tim3⁺Lag3⁺ (**G**, *n* = 5 mice) CD8⁺ T cells. Two-way ANOVA followed by Tukey's post-test in (**E**, **F**, **G**); two-tailed Mann-Whitney *U* test in (**D**). Data are plotted as mean ± SEM, **E–G** represent one out of two independent experiments.

with CTV labeled OVA-specific T cells (OT-I) for 48 h. Interestingly, cDC1 isolated from DC-therapy-treated lungs induced significantly higher CD8 T cell proliferation than cDC1 from the isotype group, especially at the lowest peptide concentrations (Fig. 7K). Moreover, the level of IFN-γ released by CD8 T cells stimulated by therapy-treated

lungs cDC1 was significantly higher than isotype (Fig. 7I–K). Collectively these data demonstrate that DC-therapy expands pre-cDC1 and cDC1 with heightened metabolism, dampens regulatory/inhibitory genes and induces rescue of immunostimulatory properties in lung resident cDC1.

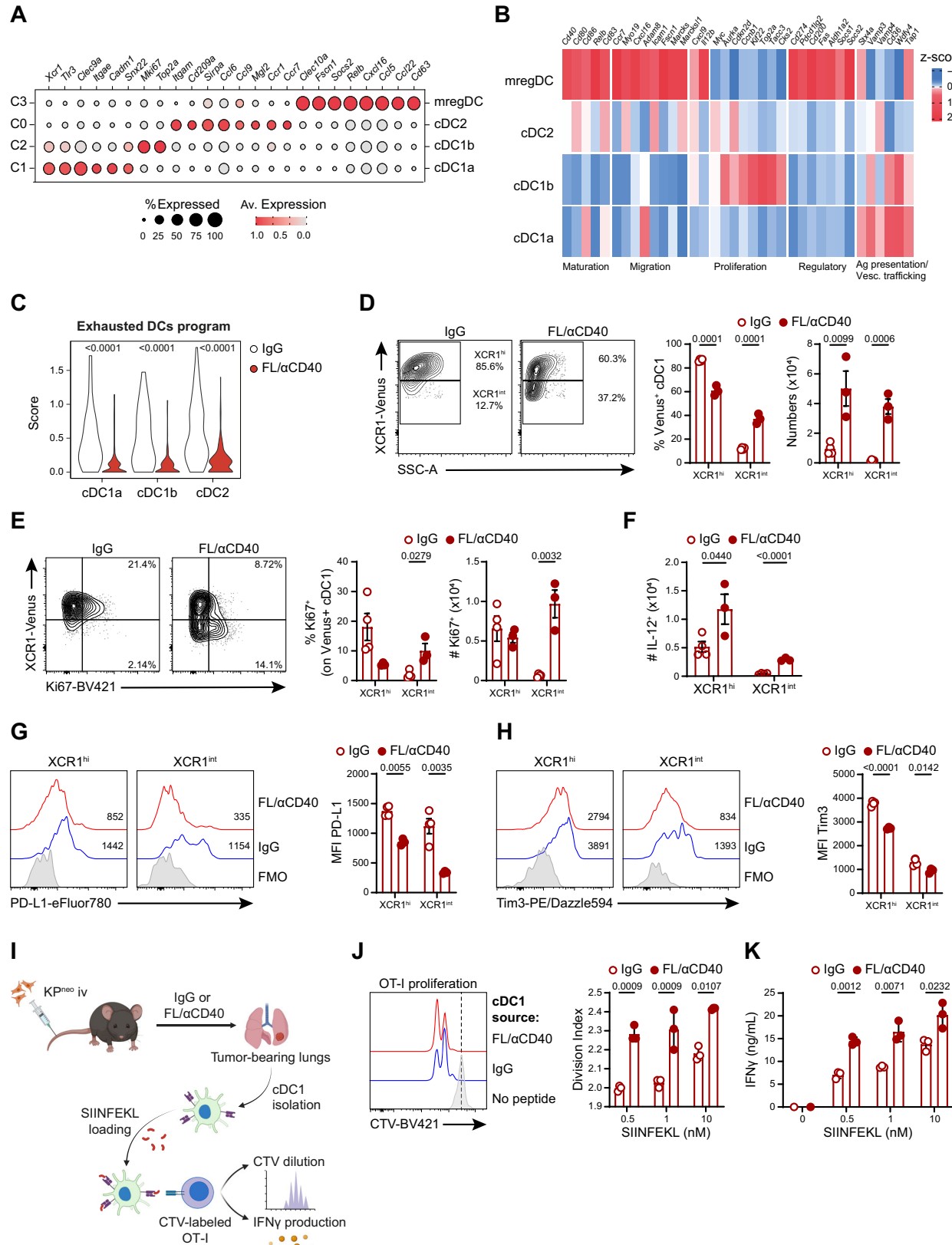

## Discussion

Evidence from clinical studies suggests that tumors with high mutational load are more responsive to checkpoint inhibitors[4,50], illustrating that neoAgs can elicit anti-tumoral immunity. Nevertheless, neoAgs presence is not decisive by itself. Thus, understanding how immunity to naturally occurring tumor antigens is initiated and regulated is critical to improve immunotherapy. In this study, we generated a model of lung cancer expressing a defined pattern of endogenous neoAgs to investigate the interplay between cross-presenting cDC1 and anti-tumoral CD8+ T cell responses. We show that cDC1 control the amplitude and diversity of CD8+ T cell responses to bona-fide neoepitopes and DCs therapeutic targeting enhances anti-tumoral

**Fig. 7 | DC-therapy induces expansion and functional rescue of cross-presenting cDC1 in lung tissues.** KP^neo tumors were implanted orthotopically in the lung and treated with FL/αCD40 or IgG as depicted in 5 A. CD45+ cells were isolated at day 9 after tumor challenge and analyzed by scRNA-seq. **A** Dot plot showing scaled gene expression of selected genes defining DC clusters. **B** Heatmap showing the normalized expression of selected genes across different DCs clusters and the association of individual genes to the indicated processes. **C** Violin plots show combined mean expression values for the indicated genes (score) for the exhaustion markers (*Axl, Ccl19, Ccl22, Aldh1*) in cells of the indicated clusters in FL/αCD40 *vs.* IgG. **A**−**C** scRNAseq data correspond to a pool of 4 animals per group. **D** Representative FC plots and quantification of the frequencies and absolute numbers of Venus+ cDC1 cells. **E** Representative FC plots and quantification of the frequencies and absolute numbers of Ki67+ cDC1 cells. **F** Absolute numbers of IL-12+

cDC1 cells in KP^neo treated with FL/αCD40 or IgG. **G, H** Representative histograms and quantification of PD-L1 or Tim3 expression in cDC1 subsets, respectively. **D**−**H** For IgG group *n* = 4 mice, for FL/αCD40 *n* = 3 mice, one out of two independent experiments. **I** Scheme describing cross-presentation assay, where sorted cDC1 from KP^neo tumor-bearing lungs were pulsed with SIINFEKL ex vivo and co-cultured with CTV-labeled OT-I T cells (Created with BioRender.com). **J** Representative histograms and quantification of OT-I proliferation. **K** IFN-γ production by OT-I after 48 h of co-culture. **J, K** *n* = 3, data represent experimental replicates from pooled sorted cDC1 from 3 mice per group, per experiment, one out of two independent experiments. two-tailed Mann-Whitney *U* test in (**C**); two-tailed Multiple *t* tests in (**D**−**K**). Data are plotted as mean ± SEM, represent one out of two independent experiments.

responses qualitatively and quantitatively. While the role of cDC1 in enhancing linear responses to individual model antigens is well accepted, this study proves that cDC1 broaden responses to multiple neoAgs generated by non-synonymous mutations, including poorly immunogenic ones. This proof-of-concept study indicates that cDC1 boosting may be particularly beneficial in heterogenous tumors expressing a range of immunogenically diverse neoAgs.

Deletion of the DNA repair gene *Mlh1* in the poorly immunogenic parental KP model enhanced the tumor mutational load and generated predicted MHC class-I neoepitopes, as previously achieved in other cancer types and autochthonous KP[32,36]. Further on, we identified and tested predicted neoAgs commonly expressed by the parental line and the hypermutated line and those accumulating exclusively in MLH1 deficient tumor cells. Intriguingly, shared epitopes in KP control were not immunogenic in vivo but elicited responses when expressed together with *de-novo* generated immunogenic neoAgs in KP mutated cells. These data align with recent reports showing that a relatively strong model antigen can promote immunogenicity of a second weaker antigen[51,52], extending this notion to the context of genuine multiple bona-fide neoAgs. Of note, the pattern of responding neoAgs was modified by cDC1 depletion or, conversely, by cDCs expansion, suggesting that weaker neoAgs (low pMHC-I affinity or low expression) may preferentially require boosting of cross-presenting cDC1. In line with this concept, recent findings showed that DCs can shape the final repertoire of anti-tumoral T cells[53].

The KP^neo model allowed us to formally explore the relation between neoAgs density and response to therapy. Increasing neoAgs content in lung tumors was not sufficient to confer responsiveness to PD-L1 checkpoint blockade, recapitulating what occurs in a fraction of NSCLC patients. PD-L1 blockade was shown to rescue T cell responses to immunogenic MC38 via de-repression of cDC1 that express high PD-L1[54]. In KP this is not sufficient, indicating that lung tumor may express further inhibitory axis. This is in line with other reports documenting the refractoriness of KP to various combinations of ICB, even in the presence of strong artificial antigens[19,21,32,33]. In contrast, Flt3L-mediated expansion of the cDCs compartment combined to an immunostimulatory molecule (αCD40), in the absence of T cell targeting, had a major impact and inhibited tumor growth, proportionally to the immunogenicity and density of neoAgs. This result is consistent with findings in pancreatic tumor models where radiation therapy to increase neoAgs synergises with a combination of Flt3L and αCD40[24,55].

Multiple recent reports highlight the critical role of local tissue factors for imprinting anti-tumoral responses, questioning the translational value of heterotopic cancer models[56]. The native microenvironment of lung tumors affects resident cDC1[17,21,57] and it is hostile to anti-tumoral effector CD8+ T cells[19], providing a strong rationale for the use of Flt3L, as currently being tested in clinical trials (NCT04491084). Indeed, a recent study showed that Flt3L + αCD40 can promote a reservoir of functional CD8+ T cells to the model antigen OVA in lung tumors draining lymph nodes[34]. Our results confirm these

previous data in the setting of multiple and diverse neoAgs. Lung tumors accumulate heterogenous lymphocyte states encompassing early precursors, memory like precursors and late exhausted/dysfunctional cells, whose functional significance and responses to ICB are under intense investigation[58,59]. scRNA-seq data in our model broadly identifies these diverse CD8 T cell states indicating that DC-therapy induces a burst in proliferation and upregulation of effector genes (*Gzma, Gzmb, Klrg1*) while reducing exhaustion, across all non-naïve CD8+ T cells subsets. These results, combined to the pattern of response to neoAgs, led us to speculate that promoting cross-presentation may help to engage novel T cell specificities, bypassing exhausted T cells directed to dominant tumor antigens. Our data are consistent with an impact of DC-therapy on cross-presentation, since responses are strongly reduced in cDC1 deficient hosts, ruling out bystander effects of Flt3L on other immune subsets. We foresee that increasing the mutational load may have promoted also the generation of MHC class-II antigens, although we have not explored this aspect extensively. Consistently, number of CD4 T cells infiltrating KP^neo in basal conditions were higher and we have observed transcriptional changes in CD4 T cell clusters after therapy. In light of the increasing role of CD4 T cells and the crosstalk between CD4 and CD8 during anti-tumoral T cell responses[60–62], the KP^neo model and its MHC class-I loss variant represent a useful tool to address the interplay of natural class-I and -II neoAgs.

Focusing on the DCs compartment, we found that therapy expands preferentially a subset of immature cells enriched in histones and cyclins (cDC1b), consistent with their identity as replicating committed pre-cDC1[45] and partially expands fully differentiated cDC1, while having less impact on cDC2[63]. Gene set enrichment analysis confirmed enhanced mitochondrial respiration and RNA metabolism (RNA splicing, ribosome biogenesis, t-RNA), indicative of an increased metabolism and translation rate. In contrast, the cluster corresponding to mReg was strongly reduced, likely reflecting enhanced mobilization to tumor draining lymph nodes. Although we have not examined DC-therapy-driven molecular changes in the DCs compartment in KP^ctrl tumors extensively, flow cytometry validation of some parameters suggests that DC remodelling is antigen-agnostic. Intriguingly, multiple genes defining regulatory and inhibitory pathways were significantly downmodulated in DCs subsets post-therapy, suggesting a rejuvenated DCs compartment, not yet skewed by the lung environment of lung tumors. Accordingly, cDC1 isolated from therapy-treated lungs were more immunostimulatory than the untreated counterparts, demonstrating cell autonomous functional rescue. In summary, our findings demonstrate that anti-tumoral responses against endogenous neoAgs in mutated lung tumors are critically regulated by cross-presenting cDC, impacting on the diversity and functionality of tumor-reactive CD8 T cells. Moreover, these results suggest that therapies to expand and remodel the cDCs compartment can bypass checkpoint refractory CD8+ T cells, by engaging novel effectors with a broad range of specificities.

## Methods

### Mice

Mice were housed at the ICGEB animal house facility. Wild type C57BL/6 JOlahsd were purchased from ENVIGO and kept in our facility under inbreeding conditions. Batf3[−/−] mice were kindly donated by Dr. Christian Lehmann (Erlangen university hospital) and maintained in our facility on a pure C57BL/6J background. XCR1-Venus mice were generated as described[38] and kindly provided by Professor Kastenmuller, (Wurzburg Institute of system Immunology). OT-I (C57BL/6-Tg (TcraTcrb)1100Mjb/J) mice were purchased from Jackson Laboratories and hosted in our animal facility. The study was approved by International Centre for Genetic Engineering and Biotechnology (ICGEB), board for animal welfare and authorized by the Italian Ministry of Health (approval number 370/2021-PR, issued on 28/05/2021). Animal care and treatment were conducted with national and international laws and policies (European Economic Community Council Directive 86/609; OJL 358; December 12, 1987). All experiments were performed in accordance with the Federation of European Laboratory Animal Science Association (FELASA) guidelines institutional guidelines and the Italian law. Mice were maintained at the ICGEB animal Bioexperimentation facility in sterile isolators (12 h/12 h light/dark cycle, T 21 °C ± 2 °C, RH 55% ± 10%) and received a standard chow diet and water *ad libitum*. During this study, females 8–12 weeks old were used and animals were randomized among groups. No statistical methods were applied a priori to calculate group sample size, all experiments were performed at least twice with a minimum of 3 animals per group. Experiments with mice were terminated based on humane endpoint criteria approved by our institutional board; fixed time points of analysis are indicated in the figure legends. For s.c. tumors, ulceration signs were considered as endpoint criteria, and for orthotopic tumors 10% weight loss was used as endpoint criteria. Tumor burden scores and tissue fluorescence were assessed blindly. All data were included unless there were technical issues with experimental setup or data collection.

### Tumor cell lines

The KP line has been isolated from primary lung tumors of C57BL/6 KP mice (K-ras[LSLG12D/+]; p53[fl/fl] mice)[35]. The line was kindly provided by Dr. Tyler Jacks (Massachusetts Institute of Technology, Cambridge, USA), and used previously[17]. Mlh1 knockout clones were generated by Crispr Cas9 based technology using 2 single guide RNAs (sgRNA) targeting Mlh1 exon 5 as described in ref. [36]. Briefly, KP cells were transiently transfected with pZac2.1-U6sgRNA-CMV-ZsGreen and (pSpCas9(BB) (PX458) using Lipofectamine 3000 (Invitrogen), following the manufacturer´s instructions. ZsGreen[+] cells were cell sorted and single clones were tested for Mlh1 expression. KP[ctrl] cells derive from cells transiently transfected with only (pSpCas9(BB)(PX458). MHC-I expression was evaluated by FC after stimulation with 0.2 ng recombinant IFNγ.

### Identification of neoAgs expressed in KP lines

Genomic DNA (gDNA) was extracted from C57 spleen and KP, KP[ctrl] and KP KO cell lines using DNA Blood and cell culture kit (Qiagen) following the manufacturer´s instructions. Whole-exome sequencing (WES) was performed at Macrogen Europe as follow: Next Generation Sequencing (NGS) libraries were prepared using SureSelectXT Library Prep Kit and sequenced on Illumina platform as paired-end 151 bp reads. FastQ files provided by Macrogen Europe were analyzed using a bioinformatics pipeline previously published[64]. On average, we obtained 99% of the exome region covered by at least one read and a median depth of 98x on sequencing data. Mutational calling was performed using the comparison strategy: mouse germline alterations were subtracted by using normal DNA sequenced from the spleen of C57 mice. We selected only variants supported by 5 or more altered reads. For bulk RNA-seq of tumor cell lines, RNA was extracted from KP, KP[ctrl] and KP[neo] cells

using RNeasy Mini kit (Qiagen) (3 samples/condition) and sent to Macrogen Europe that generated sequencing data using the TruSeq Stranded mRNA LT Sample Prep Kit for generating NGS library which were sequenced on Illumina platform as paired-end 101 bp reads. FastQ files produced by Macrogen Europe were analyzed as previous published[65]. In brief, data was aligned using MapSplice2[66] and *mm10* assembly as reference genome. The generated BAM files were post-processed to translate genomic to transcriptomic coordinates and used as input to RSEM[67] for gene expression quantification employing GENCODE vM9 as transcript annotation. The neoantigen prediction analysis was performed using a previous published bioinformatic pipeline[36] setting C57 mouse haplotypes: H2-Kb and H2-Db. Briefly, the mutational calling data, i.e., SNVs and indels, was used for generating mutated peptides of length 8–11 and then employed as input of NetMHC 4.0 software. Only peptides with predicted strong binding affinity (%rank ≤ 0.5) were considered. To further filter out predicted neoantigens based on expression level we performed the mutational calling also on the BAM files generated from RNA sequencing data. We set 3 as the minimum number of altered allele and matched the results with those obtained by WES. GSEA was used to compare KP[ctrl] and KP[neo] cell lines. Log$_2$FC-ordered gene list was used to retrieve gene set enrichment using the fgsea package in R (v. 1.26.0). Gene sets representing relevant biological processes were selected from Reactome and Hallmarks from MsigDB database.

**In vitro cell line proliferation rate.** To test the proliferation rate of the edited cells, crystal violet (Sigma) assay was performed. For this, $3 \times 10^3$ cells from KP, KP[ctrl] and KP KO clones were seeded in 96-well plates and kept for 6, 24, and 48 h. Cells were fixed with 1% paraformaldehyde and stained with 0.5% crystal violet for 20 min. Cells were washed, and the dye was solubilized by adding 1% sodium dodecyl sulphate (Sigma-Aldrich). Optical density was measured at 570 nm using microplate reader (Biorad). Proliferation rate was calculated as fold change of the optical density found after 6 h, consider as day 0.

**Western Blot.** Cellular proteins were obtained by whole cell lysis using radioimmunoprecipitation assay buffer (50 mM Tris HCl, 150 mM NaCl, 1.0% (v/v) NP-40, 0.5% (w/v) Sodium Deoxycholate, 1.0 mM EDTA, 0.1% (w/v) SDS and 0.01% (w/v) sodium azide at a pH of 7.4), containing proteinase inhibitor (Sigma-Aldrich). Protein concentration was measured by BCA assay (Pierce). Proteins from KP, KP[ctrl] and KP KO lysates were analysed by SDS-PAGE. Membranes were blocked with 3% BSA-TBS for 2 h and were incubated with an appropriate primary antibody, overnight at 4 °C. Primary antibodies includes anti-MLH1 (clone EPR3894, Abcam, dilution 1:1000) and anti-Tubulin (clone 11H10, Cell signalling, dilution 1:1000). After rinsing, the membranes were incubated with HRP-conjugated secondary antibodies (Goat anti-mouse IgG, Invitrogen, dilution 1:5000 or Goat anti-rabbit IgG (H + L), Invitrogen, dilution 1:1000) for 45 min. Finally, membranes with incubated with ECL (Biorad), chemiluminescent signal was developed and imaged by ChemiDoc.

### Tumor outgrowth

To establish subcutaneous tumors, $5 \times 10^5$ cells were injected in the right flank of 8–12 weeks old female mice. For the orthotopic model, $5 \times 10^4$ cells were injected intravenously. To assess growth in the s.c. model, measurements were collected at day 0, 10, 12, 15, 18 and 21 using a caliper. Mice carrying s.c. tumors were sacrificed at day 12 or 21. Tumor size was calculated as $V = \frac{d^2 \times D}{2}$ (where, d = minor tumor axis and D = major tumor axis) and reported as tumor mass volume (mm³ of individual tumor volume). Lungs were harvested at day 9 as indicated in the legends. To evaluate tumor burden in lung tissues, organs were harvested, fixed in formaldehyde 10% and paraffin embedded following the standard procedure. Consecutive sections every 200 µm were dewaxed and rehydrated and stained with the H&E (Bio-Optica, Milano

Spa). Ilastik (v. 1.3.2) software was trained to automatically detect and segment tumor nodules. The area of tumor nodules was quantified over consecutive sections and averaged (4 sections/sample). Measurements and automatic thresholding were performed using Fiji (v. 1.53c). The area occupied by tumor nodules was expressed as a function of the total lung lobe area.

## Mouse experimental manipulation

For CD8 depletion, mice were injected intraperitoneally (i.p.) with 200 µg of α-CD8 (clone YTS169.4) or rat IgG2a control antibody (clone 2A3) at day 3, 6, 9, 12 and 15 post tumor implantation. Tumor tissues (s.c.) were harvested after 21 days. For checkpoint blockade in s.c. setting, KP$^{ctrl}$ or KP$^{neo}$ challenged mice were treated i.p. with 200 µg of α-PD-L1 (clone 10 F.9G2) or rat IgG2a isotype control antibody at day 3, 6, 9 and 12 post tumor inoculation and tissue harvested at day 21. For treatment of KP$^{ctrl}$ or KP$^{neo}$ orthotopic tumors mice were treated i.p. with 200 µg of α-PD-L1 or rat IgG2a isotype control antibody at day 5 and 7 post tumor inoculation and tissue harvested at day 9. DC-therapy included 30 µg of rhuFLT3L (Celldex) and 80 µg of α-CD40 (clone FGK4.5FGK45), or 80 µg of rat IgG2 (clone 2A3), for all experimental groups. For treatment of KP$^{ctrl}$ or KP$^{neo}$ tumors implanted subcutaneously, rhuFLT3L was administered i.p. at day 3, 5, 7, 9 and 11; aCD40 was administrated i.p. at day 7 and 11 and tissues were harvested at day 21. For earlier time point (day12), mice were treated i.p. with rhuFLT3L at day 3, 6 and 9 and aCD40 was administrated i.p. at day 8. For treatment of KP$^{ctrl}$ or KP$^{neo}$ orthotopic tumors rhuFLT3L was administered i.p. at day −2, 2, 4 and 6; and α−CD40 was administered i.p. at day 5. Tissues were harvested at day 9.

## Tissue labelling

Lungs were harvested after perfusion and fixed in 10% formaldehyde and paraffin-embedded following the standard procedure. 5 µm sections were treated with antigen-retrieval solution (Vector laboratories), neutralized in $H_2O_2$ to block endogenous peroxidase and blocked in 10% goat serum, 0.1% Tween. Slices were incubated overnight in a humidified chamber in a 1:200 dilution of anti-mouse CD8 (Invitrogen cat 14-0808-82) in blocking buffer. Detection was performed using the ImmPRESS polymer detection system (Vector Laboratories), according to manufacturer's instructions. CD8$^+$ T cells quantification was done using ImageJ software. For immunofluorescence of tissues, s.c. tumors were harvested and fixed in 4% paraformaldehyde (PFA) and cryo-preserved in 30% sucrose in PBS overnight. Lungs were intratracheally perfused with 1% PFA and fixed in 4% PFA and cryopreserved in 30% sucrose in PBS overnight. Tissues were embedded in a frozen tissue matrix following standard procedure and sectioned in 5 µm sections in a cryostat. For CD8 detection, sections were blocked with 5% mouse serum and labelled using anti-mouse CD8 (clone 4SM15, Invitrogen, dilution 1:800) followed by Goat anti-rat IgG (H + L) Alexa fluor 555 (Invitrogen, dilution 1:500). Images were acquired with Leica DFC450 C microscope using 20x objective. XCR1-Venus was directly detected by primary Venus fluorescence. Nuclei were counterstained with DAPI. Images were acquired with a Confocal Zeiss LSM 880 (40x objective), using ZEN Black Imaging Software (ZEISS) (v. 14.0.27.201).

## Flow cytometry

For cell staining, s.c. tumors were surgically excised, and lungs were collected after cardiac perfusion with PBS. Tissues were mechanically dissociated and digested using collagenase type-II (Worthington) and DNAse-I (Roche). Samples were filtered through a 100 µm filter to obtain a single cell suspension. Lymph nodes (mediastinal or inguinal) were smashed and filtered through a 40 µm cell strainer. FcR binding sites were blocked by using αCD16/CD32 (BioLegend) and cell viability was assessed by staining with Live/Dead Fixable Aqua Staining (Life technologies). Cells were resuspended in antibody-containing staining buffer (list of antibodies in Supplementary Table 2). Cells were fixed

with 1% paraformaldehyde. For IFNγ staining, single-cell suspensions were stimulated with cell activation cocktail (including phorbol 12-myristate 13-acetate and Ionomycin) (BioLegend) in the presence of Golgi Stop (monensin, BD Biosciences). Upon extracellular staining, cells were fixed and permeabilized using Cytofix/Cytoperm solution (BD Biosciences) following manufacturer's instructions. For Ki67, GzmB, TCF-1 and FoxP3 detection, upon extracellular staining cells were fixed and permeabilized using eBioscience FoxP3/Trascription factor Staining Buffer set (Invitrogen) following the manufacturer's instruction. For Sh1-dextramer staining in tumor tissues, custom dextramers were generated using the U-load Dextramer kit (Immudex). Dextramer staining was performed according to the manufacturer's instructions. Briefly, cell suspension was labelled with the custom dextramers for 10 min at room temperature. Absolute cell count was performed by adding TrueCount Beads (BioLegend) to the samples following manufacturer's instructions. Flow data were acquired with FACS Celesta or LSRFortessa X-20 (BD Biosciences) using BD FACSDiva Software (v. 8.0.1) and analyzed FlowJo software (Tree Star, Inc. version 10.8.2). For cell sorting for scRNAseq experiment, single cells suspensions were incubated with α-CD45-APC Fire as described before. For cross-presentation assay, total CD11c$^+$ cells were isolated from tumor-bearing lungs using CD11c microbeads (Miltenyi Biotec) and Venus$^+$ cells were sorted. CD45$^+$ and Venus$^+$ cells were isolated using ARIA Cell Sorter (BD) and number of cells were counted with trypan blue before processing for experiments.

## Ex-vivo priming assays

Sorted lung cDC1 were plated in tissue-treated 96-well plates (5,000 cDC1 per well) in complete IMDM for 3 h at 37 °C, in the presence of SIINFEKL at different concentrations (0; 0.5; 1 and 10 nM) and 1 µg/ml poly(I:C). After cDC1 were washed, 50,000 naïve CellTrace Violet (CTV)-labelled OT-I cells (5 µM CTV, Invitrogen) were added to each well in complete RPMI media and kept at 37 °C for 48 h. Aliquots of conditioned media were collected for measuring IFNγ by ELISA (ELISA MAX, BioLegend). Cells were washed and stained with anti-mouse CD3ε-FITC (BioLegend, dilution 1:200) and CD8-APC (BioLegend, 1:200), in FACS buffer for 30 min at 4 °C. Afterwards, cells were washed with PBS and stained with LIVE/DEAD Fixable Aqua Dead Cell Stain Kit (Invitrogen) according to manufacturer's instructions. Finally, cells were fixed in FACS-Fix buffer (PBS, 1% PFA), and CTV dilution was analyzed in FACS Celesta Cell Analyzer (BD). Data generated were further analyzed in FlowJo software (v. 10.8.2 version).

## CD8$^+$ T cell isolation

Single cells suspension from tumour-bearing lungs described previously were used for CD8$^+$ T cells sorting using immunomagnetic sorting using CD8α$^+$ T cell isolation kit (Miltenyi Biotec) following the manufacturer's instruction. Purity was checked by flow cytometry. Isolated CD8$^+$ T cells were resuspended complete medium and used in ELISpot assay.

## Gene expression analysis by RT-qPCR

S.c. tumors were disrupted and homogenized in Trizol. RNA extraction was performed with Trizol (Gibco) following the manufacturer's instructions. cDNA synthesis was performed using SuperscriptII and random hexamers (Invitrogen), and gene expression was determined by qPCR using EVA Green (Biorad) according with the manufacturer's instruction (specific primers are listed in Supplementary Table 3). The levels of gene expression were normalized using the expression of Gapdh or Gusb as housekeeping genes.

## ELISpot to determine neoantigen- specific T cell response

Bone marrow dendritic cells (BMDCs) were loaded with 2 µM of each peptide peptides (Supplementary Table 1) in the presence of 0.1 µg/mL LPS for three hours. $1 \times 10^4$ loaded-BMDCs were co-cultured with $1 \times 10^5$ splenic-CD8$^+$ T cells from KP$^{ctrl-}$ or KP$^{neo-}$ tumor bearing mice treated

with FL/αCD40 or IgG. ELISpot plates were coated with capture α-IFNγ (clone R4-6A2, BioLegend, 4 μg/mL) and subsequently blocked. Peptide-loaded BMDCs and CD8⁺ T cells were plated and gently mixed before incubation for 40 h. After incubation, cells were removed by washing and the plates were washed and probed with biotin α-IFNγ (clone XMG1.2, BioLegend, 2 μg/mL) followed by incubation with Streptavidin-ALP (Mabtech). IFNγ spots were developed using BCIP/NBT plus (Mabtech).

## Analysis of human data set collection
Clinical and genomics data for TCGA-LUAD PanCancer Atlas cohort were retrieved from cBioPortal on 10th January 2023 and analysed in R (version 4.2.1). Patients ID were filtered to keep only sample for which both clinical, mutational (TMB) and gene expression data were available ($n = 505$). mRNA expression of genes was retrieved as logRNAseq V2 RSEM mRNA expression Z-scores relative to all samples. Average expression of genes signature was adopted as effector CD8 and cDC1 scores. Genes defining effector CD8 T cells scores include: *Cd8a*; *Cd8b*; *IFNγ* and *Prf.1*. The signature for the cDC1 score includes *Batf3*; *Irf8*; *Thbd*; *Clec9a, Xcr1*. Signatures were derived from ref. 10. To identify two groups with high or low cDC1 score, we classified TCGA LUAD patients as the 25th bottom quartile (bottom) and the 25th top quartile (top). Correlation analysis was performed by Pearson correlation coefficient (r) using GraphPad (v. 8.4.3) (v. 8.4.3). Kaplan-Meier survival curves were generated using GraphPad (v. 8.4.3) and compared using the Log-rank test (Mantel-Cox).

## Single-cell sequencing
**Sample preparation.** Lungs from tumor-bearing mice treated with either FLT3L and αCD40, or rat IgG2a control isotype were harvested 9 days after tumor injection. Single cell suspension was obtained using gentle MACS dissociator (Milteny), filtered using 70 μm cell strainer. Samples were incubated with α-CD45-APCFire and Live/Dead Fixable Aqua Staining (Life technologies) to asses cell viability. CD45⁺ cells were isolated using ARIA Cell Sorter (BD). Single cells were prepared using Chromium Next GEM Single Cell 3′ Reagent Kits v3.1 (Dual Index) (10X Genomics), following the manufacturer's instructions. In brief, 10,000 cells/channel were partitioned on Chromium Next GEM Chip G using the Controller platform (10X Genomics). After encapsulation and simultaneous cell barcoding, library preparation was performed. Following quality controls, libraries were sequenced according to manufacturer's instruction on Novaseq6000 platform (Illumina). Target coverage was 50,000 cluster/cell.

**scRNA-seq data processing and analysis.** After demultiplexing, fastq files were processed employing the Cell Ranger (v 4.0.0)[68] workflow using default parameters. Reads alignment was performed to reference genome mm10 (reference version 2020-A, 10X Genomics). Gene expression matrices, containing the number of unique molecular identifiers (UMIs) for every cell and gene, were computed retaining only confidently mapped reads, non-PCR duplicates, with valid barcodes and UMIs. All downstream analyses were carried out in R environment (v 4.0.3) by the Seurat package (v 4.0.3) (https://satijalab.org/seurat/). Gene counts were imported as a Seurat object, filtering out genes expressed in less than 3 cells. For each sample, putative doublets were identified using scDblFinder R package (v 1.4.0)[69], setting the expected doublet rate at 5%, following 10X Genomics estimates according to the number of loaded cells. Sample count matrices were then merged and cells expressing less than 200 genes, or less than 1000 unique gene counts were discarded, along with cells with a ratio of mitochondrial versus endogenous genes expression exceeding 10% and with doublets. Raw expression data were normalized applying log2 transformation with NormalizeData function and scaled using ScaleData function, regressing on the percentage of mitochondrial gene expression and cell cycle scores, previously computed using CellCycleScoring function. Top 2000 genes

with the highest standardized variance were computed using FindVariableFeatures function (selection.method = "vst"). Principal component analysis (PCA) was computed using RunPCA function with default parameters. PCA embedding was corrected for sample batch were through the Seurat Wrapper package (v 0.3.0). When analyzing the whole immune compartment, batch effect was removed by matching mutual nearest neighbor (MNN) algorithm[70], implemented with the RunFastMNN function using default parameters. Multiscale differential abundance scores were computed using the DA-seq algorithm[71], taking as input MNN-corrected principal components. For the analysis of dendritic cells and T cells, batch correction was achieved with the Harmony algorithm (v 0.1.0)[72], by running the RunHarmony function on the first 20 PCA dimensions and theta=2. Shared Nearest Neighbor (SNN) graph was computed using the FindNeighbors function, taking as input the first 20 corrected PCA dimensions. Cell clusters were defined using Louvain algorithm with the FindCluster function. For visualization in 2 dimensions uniform manifold approximation and projection (UMAP)[73] was used. Cluster-specific genes were identified using FindAllMarkers function with options only.pos = TRUE, pseudocount.use=0.1 and logfc.threshold=1, setting a cut-off of FDR < 0.01. Differentially expressed genes upon DC therapy (FL/αCD40 versus aIgG comparison) were computed using FindMarkers function with options only.pos = FALSE, pseudocount.use = 0.1 and logfc.threshold=1, setting a cut-off of FDR < 0.01. In the violin plots, we report the mean of expression values of the selected gene signatures. Specifically, we subset the normalized gene expression matrix on the selected genes and compute the mean expression of the signature for each cell using the colMeans() function.

**Gene-set enrichment analysis.** To compare clusters upon DC therapy, genes expressed in at least 10% of cells for each cluster were ranked by decreasing order of avg_log2FC in FL/αCD40 versus anti-IgG comparison. Gene-set enrichment analysis for biological processes Gene Ontology terms was performed using the gseGO function from clusterProfiler (v3.18.1) (package, with number of permutations set to 100'000 and considering significance for *q*-values < 0.05.

## Statistical analysis
All graphs and statistical analyses were conducted using GraphPad (v. 8.4.3) Prism 8 (GraphPad (v. 8.4.3)) and R packages. All data represent mean ± SEM. The number of biological replicates and statistical test are indicated in the figure legend. Comparison between two groups was tested by parametric Student t-test or Mann-Whitney *U* test. For multiple comparison, parametric one-way analysis of variance (ANOVA) was performed followed by Tukey's posttest. When two variables were analysed among two or more groups two-ways ANOVA was performed, followed by Sidak's posttest, only during the in vitro growth of KP KO clones Tukey's posttest was used. *$p < 0.05$, **$p < 0.01$, ***$p < 0.001$, ****$p < 0.0001$.

## Reporting summary
Further information on research design is available in the Nature Portfolio Reporting Summary linked to this article.

# Data availability
Single-cell RNA-seq data have been deposited at ArrayExpress data repository under the accession number E-MTAB-12508. Whole exome sequencing data have been deposited at Sequence Read Archive (SRA) data repository under the accession number PRJNA1082088. RNA-seq data have been deposited at Gene Expression Omnibus (GEO) data repository under the accession de number GSE260743. The TCGA publicly available data used in this study are available in the cBioPortal database https://www.cbioportal.org/study/summary?id=luad_tcga. The remaining data are available within the Article, Supplementary Information or Source Data file. Source data are provided with this paper.

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

## Acknowledgements

This work was supported by AIRC IG 21636 to F.B. L.L., L.G.M. and S.J. were supported by ICGEB pre and post-doctoral fellowships. RA is supported by Italian Telethon. N.C. received support by fellowships from Fondazione Umberto Veronesi (FUV). P.B. was supported by the Fondation pour la recherche sur le cancer French fellowship (ARC). A.B. was supported by AIRC IG 2023-28922 and AIRC IG 2018-21923. G.G. was supported by AIRC MFAG 2020-24604. We also acknowledge Dr. Fulvia Vascotto from TRON for discussing data and providing insights.

## Author contributions

F.B. and L.L. conceived the study. F.B. supervised planning and execution of experimental plan. L.L., L.G.M., P.B., S.J., G.M.P., S.V., C.V. performed research and analysis. F.G. and D.L. performed scRNAseq library preparation and analysis. P.S., G.G., A.B., M.D. provided useful reagents. VR and LP performed dextramer staining experiments. L.L., P.B., P.G., M.D., N.C., P.S., contributed to data interpretation and manuscript writing. L.L. and L.G.M. prepared figures. F.L.T., R.A., G.R., performed bioinformatic analysis.

## Competing interests

A.B. and G.G. are cofounders and shareholders of NeoPhore. A.B. is a member of the NeoPhore scientific advisory board. The remaining authors declare no competing interests.

## Additional information

[1]Cellular Immunology, International Centre for Genetic Engineering and Biotechnology, ICGEB Trieste, Italy. [2]San Raffaele Telethon Institute for Gene Therapy (SR-Tiget), IRCCS San Raffaele Scientific Institute, Milan, Italy. [3]Université Paris Cité, Institut Cochin, INSERM 1016, Paris, France. [4]Department of Oncology, Molecular Biotechnology Center, University of Torino, Turin, Italy. [5]G. Armenise-Harvard Immune Regulation Unit, IIGM, Candiolo, TO, Italy. [6]Candiolo Cancer Institute, FPO-IRCCS, Candiolo, TO, Italy. [7]Center for Omics Sciences, IRCCS San Raffaele Institute, Milano, Italy. [8]Vita-Salute San Raffaele University, Milan, Italy. [9]IFOM ETS - The AIRC Institute of Molecular Oncology, 20139 Milan, Italy. [10]Department of Dermatology, Venereology & Allergology, Medical University of Innsbruck, Innsbruck, Austria. [11]Aix-Marseille University, CNRS, INSERM, CIML, Centre d'Immunologie de Marseille-Luminy, Turing Center for Living Systems, Marseille, France. [12]Institut Pasteur, CNRS 3738, University de Paris Cité, Paris, France. [13]Present address: Laboratory of Tumor Inflammation and Angiogenesis, Center for Cancer Biology, VIB, KU Leuven Leuven, Belgium. [14]Present address: Boehringer Ingelheim RCV GmbH & Co KG, Vienna, Austria. [15]Present address: Cellular and Molecular Oncoimmunology, IRCCS Humanitas Research Hospital, Rozzano, Italy. [16]Present address: Department of Bio-medical Sciences, Humanitas University, Pieve Emanuele, Italy. ✉e-mail: benvenut@icgeb.org

