## [Peer Review File · Nature Communications]

Dendritic cell-targeted therapy expands CD8 T cell responses to bona-fide neoantigens in lung tumorsREVIEWERS' COMMENTS:

Reviewer #1 (Remarks to the Author): with expertise in DCs, cancer immunology

The manuscript by Lopez-Rodriguez et al investigates the immune activation triggered by one NSCLC cancer model with high TMB (KPneo, which was previously described) in mice and the efficacy of immunotherapies in controlling the grafting and outgrowth of those tumors. The authors uncover that KPneo tumors induce a notable (CD8+) T cell response compared with the parental cancer cells when grafted s.c. or orthotopically. Furthermore, KPneo tumors respond to therapies targeting/expanding DCs, but not anti-PD-L1 therapy (at least when grafted s.c.), and anti-CD40+Flt3L treatment causes a remodeling of CD8+ T cells and DCs present in the lung of KPneo tumor-bearing mice. Finally, cDC1 signatures correlate stronger with effector CD8+ T cell presence and fraction survival in TMB-high LUAD patients.

Overall, despite being based on one sole mouse tumor model, this study suggests interesting and novel findings that might be of high value for the community (which is highly appreciated): that TMB-high NSCLC tumors might be refractory to immune checkpoint blockade, but respond to DC expanding/targeting-therapies that remodel DC populations in the tumor-bearing lung. Unfortunately, both findings are poorly experimentally supported or explored, important controls and experimental groups are missing in many experiments in the manuscript, and the study is entirely biased on CD8+ T cells as DC-induced effector cells and CD4+ T cells are not considered (see detailed comments below). If those concerns are addressed, the study will represent a relevant work to be published in Nature Communications.

Major comment 1 – The responses of NSCLC with or without TMB to immune checkpoint therapy and DC-expanding/targeted therapy is poorly addressed

In my view, the differential response to those immunotherapies and a potential dependence on high-TMB probably represents the most novel and relevant finding of this study and, therefore, should be properly and systematically investigated:

In subcutaneous growth experiments (Figure 1-3), treatment with anti-PD-L1 is never compared with poly:I:C+Flt3L or aCD40+Flt3L. And, as mentioned in detail below, important controls/experimental groups are missing. To better establish the responsiveness of KPctrl and KPneo tumors to therapies, at least 2 types of immune checkpoint therapies should be tested in both models. This is especially important because anti-PD-L1 is actually (mainly) targeting cDC1s (PD-L1 is expressed by cDC1s, upregulated upon Ag uptake; and crucial for therapy efficacy, PMID: 35122038 and 32973173). Hence, testing of immune checkpoint antibodies not targeting DCs would be highly relevant, such as anti-Ctla4 or anti-PD1. While it is greatly appreciated that 2 combinations of cDC1-targeting approaches are used in the manuscript, it is entirely unclear why the authors use those in different settings. Please analyze 2 different immune checkpoint therapies and 2 different cDC1-targeting strategies in the same system (ideally in orthotopically grafted tumors) to determine if KPctrl and KPneo tumors respond to immune checkpoint blockade or DC-expanding/targeting therapy (as monotherapies and combination-treatments).

Moreover, the treatment regimen used in Figure 5 is suboptimal and largely precludes seeing effects of immune checkpoint/anti-PD-L1 therapy – which usually takes more than 3 days to have an effect – while Flt3L treatment was given for 8 days and started even before tumor injection. Hence, please administer both treatments at the same time during tumor growth (as for example done in Figure 2F) to be able fairly evaluate the effect of the different therapies (and if they work only in high-TMB tumors or not).

It is intriguing to see that CD8+ T cells in s.c. KPneo tumors and lungs harboring KPneo tumors upregulate PD-1 (at baseline or upon DC-targeted therapy). Yet, anti-PD-L1 blockade is not effective and DC-targeted therapy does not appear to have an additive effect with anti-PD-L1 therapy. Could the authors comment on this surprising finding?

Finally, the first paragraph of the results is entitled “Immunogenic KP encoding bona fide neoAgs are refractory to checkpoint inhibition”. I agree that an entire paragraph should be dedicated to those analyses (see my comments above), however as of now immune checkpoint inhibition was not explored at all in the following paragraphs.

Major comment 2 – It is not clear if the remodeling of T cells and DCs in KPneo tumor-harboring lungs depends on the high TMB burden of those tumors or the presence of tumors at all. Could that be a general feature of aCD40/Flt3L (DC-expanding/targeting) therapy?

The second most relevant finding is the notable remodeling of DC populations (mainly cDC1s) and (likely subsequently?) CD8+ T cell populations upon antiCD40+Flt3L therapy. The important remaining question is if this remodeling is unique to NSCLC with high-TMB (or tumor-presence in the lung)? Hence, could the authors investigate the most notable remodeling features (as determined in Figure 6 and 7) in cDC1s, cDC2s and CD8+ T cells (by flow cytometry and/or RT-qPCR) in naïve and KPctrl tumors-bearing mice (in the latter also the disappearance of mregDCs) treated or not with antiCD40+Flt3L therapy? This is important information to better understand the mechanisms behind the effect of this therapy (is it dependent on the presence of a cancer in the lung? Is the TMB of the cancer relevant?) and, hence, the selection of patients with cancer for this type of therapy.

Major comment 3 – Vital controls are missing in tumor growth experiments with/without therapy, the enhanced immune activation in KPneo tumors at baseline is largely omitted and tumor development on cDC1-deficient mice has to be analyzed in the absence of therapeutic treatments and of KPctrl cells.

The title of the manuscript reads “DC targeted therapy breaks immune tolerance against neoantigen in refractory lung adenocarcinoma”. However, the authors report a significant immune activation in KPneo tumors compared with KPctrl tumors in mice (Fig. 1, 2, S1 and S3) and those cancers progress much slower – so there is clearly no “immune tolerance”, but an immune induction in those high TMB KPneo cancers. Therefore, the title has to be changed (for the combination/comparison with immune checkpoint blockade, please see my major comment1). Moreover, the treatment with polyI:C+Flt3L+anti-PD-L1 or aCD40/Flt3L simply appears to amplify the pre-existing immune induction in KPneo tumors and does not appear to induce additional immune responses. However, the results in Figure 2I+J in the IgG-treated groups are not consistent with the findings of the authors in Figure

S1, where the authors already show a significant T cell-mediated immune response in mice grafted with KP-neo tumors compared with KPctrl tumors. Could the authors please explain those discrepancies?

In the s.c. grafted tumors, while it is clear that anti-PD-L1 alone does not reduce the growth of KPneo tumors, this manuscript lacks the essential control of treatment of KPctrl and KPneo tumors with polyI:C+Flt3L only (without immune checkpoint inhibition). Is this treatment causing the tumor growth difference and what is the relevance of anti-PD-L1 treatment in this setting? Also, could the authors please quantify cDC1 presence and activation (e.g. activation markers CD40, CD80, CD86) in s.c. tumors and tdLNs of KPctrl and KPneo-tumour bearing mice being untreated, treated with anti-PD-L1 monotherapy and polyI:C+Flt3L therapy (are cDC1s equally enriched and activated in KPctrl as KPneo-bearing mice and how do the treatments affect this?), especially because the authors chose an unusual intratumoral administration route for Flt3L and there is a massive size difference in examined tumors. cDC1 presence is only quantified upon aCD40/Flt3L treatment in the lung of KPneo tumor-bearing mice, however it remains unclear if anti-PD-L1 affect cDC1 presence or activation (given that those cells express this co-inhibitory molecule and might be depleted)? Also, could the authors use flow cytometry to quantify cDC1s and cDC2s (as it was done in Figure S5) in lungs and s.c. grafted tumors, as this is much more quantitative.

Furthermore, more details are needed for the choice of route of immunotherapy administration. Firstly, could the authors please explain how anti-CD40, Flt3L and anti-PD-L1 was administered to mice with orthotopic KPneo tumors in the lung (I couldn't find this information in the main text, Figure legends or the Methods) – intratumorally as well? Secondly, why was the intratumoral route chosen for polyI:C, Flt3L and anti-PD-L1 (other than Salmon et al administering polyI:C intratumorally)? Intratumoral doesn't appear to be a feasible administration route for patients with lung cancer. Moreover, is not used for Flt3L of anti-PD-L1, which are usually given intraperitoneally (for example, in the study Salmon et al, Flt3L is administered intra-peritoneally, because it has to reach the bone marrow rather than the tumor to expand cDC1s). Could the administration of anti-PD-L1 intratumorally instead of interperitoneally account for the non-responsiveness of s.c. grafted KPneo tumors? Please comment on those issues and, importantly, clarify the administration route

for the tumors grafted in the lung. Ideally, for the experiments comparing immunotherapies mentioned in major comment 1, a feasible and commonly used administration route should be chosen – additional to a treatment regimen that allows for side-by-side comparison of those immunotherapies.

Why were the experiments using Batf3-ko and XCR1-DTA separated in the manuscript? That causes confusion. Moreover, could the authors please report the tumor growth and T cell activation of KPctrl cells grafted onto those cDC1-deficient mouse models, with or without therapy? This is important to understand if the observed effects are due to the altered TMB or simply due to cDC1 loss (and hence occurring in the parental cancer as well). Also, how is tumor growth of KPneo cells on Batf3-ko and XCR1-DTA mice without therapy – is cDC1 depletion causing the effect or does it only happen upon therapy? The dramatic reduction of CD8+ T cells in untreated Batf3-ko mice grafted with KPneo tumors suggests a therapy-independent effect of cDC1 loss. Moreover, to understand the mechanism, could the authors investigate the effects of polyI:C+Flt3L independently of anti-PD-L1 on those cDC1-deficient mouse models? Those points have to be addressed before the authors can conclude that “cDC1 (...) contribute to expand the intensity and diversity of CD8+ T cell responses to neoAgs”

Finally, were tumors analyzed at a timepoint when they had a similar tumor size or at endpoint (please see rationales for that in my comments below). If not, please analyze equal-size tumors wherever possible and, otherwise, clearly state the caveat of having to investigate differently sized tumors for every experiment.

Major comment 4 – The authors unfortunately solely focus on CD8+ T cells and ignore the emerging role of CD4+ Th1 cells in anti-cancer immunity and the importance of cDC1s controlling their activation and vice versa (PMID: 32457487; 32788723 and 30961656).

The connection between cDC1s and CD4+ (Th1) T cells is strongly emerging in the literature, which should be considered by the authors in this study. Considering the importance of anti-tumor functions of CD4+ Th1 T cells in cancer and the potency of cDC1s to induce those cells

(and their cross-talk; PMID: 32457487; 32788723 and 30961656), the induction of CD4+ T cell responses and their relevance for growth of KP-Neo cancers should also be determined to better understand the experimental high TMB model that the authors use and the underlying mechanism.

Firstly, could the CD4+ T cell clusters (as shown in Figure S4B) be analyzed in-depth, as was done for CD8+ T cells in Figure 6. Does DC-expanding/targeting therapy also remodel CD4+ T cells, or is that exclusive to CD8+ T cells?

Secondly, in Figure 1-3, the authors did not formally show that it is CD8+ T cells that are specific for the neoantigens expressed by the KP-Neo cells (for example, total IFN γ was measured after re-stimulation of the entire LN). Also, according to the nature portfolio reporting summary, CD8+ T cells were gated as being CD45+ CD3+ (This information is entirely missing from the methods and should be added). Hence, could the authors re-analyze their data and focus on CD3+ CD8-negative cells (which are largely CD4+ T cells) in their respective Flow Cytometry analyses in Figure 1, 2, 3, 5D, S1, S2 and S3? Especially the number/frequency of those CD3+ CD8- T cells, their expression of CD62L, CD44, PD-1 and IFN γ in the different tumors and treatment regimens?

The addition of those analyses of CD8- T cells and the CD4+ T cell clusters of the scRNAseq will add vital information on the activities of cDC1s controlling all cancer-relevant T cell subsets in cancer (especially since cDC1s are known to induce/crosstalk with CD4+ (Th1) cells in cancer; PMID: 32457487; 32788723 and 30961656) in a TMB-high setting and will only require re-analysis of already collected data.

Finally, please state in the legend of Figure 1E at which days anti-CD8 was administered. Depleting CD8+ cells only 3 days after tumor cell injection might aide tumor engraftment rather than the out-growth of cancers (and this information should be easy-access for the reader). Hence, is tumor progression of KP-Neo tumors also accelerated when CD8+ T cells are depleted after 12 days, when the KP-ctrl tumors start to outgrow KP-neo tumors? Also, does CD8+ T cell depletion affect growth of KP-Neo tumors specifically, or in other words, how is the growth of KPctrl tumors affected after CD8+ T cell depletion?

Additional minor comments

The authors write: “Whole exome sequencing showed an increase in single nucleotide variant (SNVs) and frameshifts as compared to KP control cells (KPctrl), which corresponded to 66 and 126 predicted MHC class-I epitopes, respectively (Figure S1B).” Unfortunately, it is not well explained how the number of predicted MHC-I epitopes (and binding affinity) was determined. Please add some details to the Figure legend.

How did the authors control that KPKO2 (KPneo) cells did not accumulate further detrimental mutations affecting tumor progression (after the analysis in Figure 1) due to their deficiency in Mlh1? Especially when growing in vivo? How and why was interference with Mlh1 chosen – is it frequently mutated in NSCLC and correlative with high TMB? Please explain the rationales for this experimental approach to induce a high TMB.

Figure S1F-J should be repeated at day 12 of tumor growth – when ctrl and neo tumors have an equal size. Such an analysis is much more meaningful, because the results of the analysis performed by the authors at end stage likely reflect the difference in tumor size/progression and it is unclear if that is the cause for differential tumor growth. Only an analysis of tumors at a time before the growth difference between ctrl and neo becomes apparent will answer this question. In line, at which time point was the analysis in Figure 1F performed? It should be done (with at least a few neoantigens) at around day 12 as well. Same for Figure 2A+B, please analyze cDC1 presence in tumors at day 12 and not when the microenvironments are so dramatically different due to tumor size. Alternatively, please clearly state in the text that tumors were analyzed as such different sizes and the caveats that it poses on the interpretation of the results.

Title of Figure 7: it should read “Flt3L”. And, how do the authors know that those cDC1s are lung-resident and have not been recently recruited (especially pre-cDC1s) by the tumors? Or do the authors assume it is a specific activation of lung pre-cDC1s (as the cDC1b cluster is proliferating)?

Figure S5D – could the authors provide a complete list of deregulated genes as a

supplementary excel file?

The authors write “Altogether, expansion of pre-cDC1 and cDC1 (...) replenishes the lung with naive cDCs with increased ability to sample and present tumor antigens”. However, the authors did not actually test the ability of cDC1s for phagocytosis/AG sampling or their AG processing or presentation capacity. Related genes were also not mentioned to be deregulated in the scRNAseq analysis. Hence, could the authors please rephrase their statement or assess those functional traits of cDC1s?

Font in Figures is generally quite small and hard to read.

Reviewer #2 (Remarks to the Author): with expertise in lung immunology/DC, cancer immunology

Rodriguez et al. generated a hypermutated variant of KP lung adenocarcinoma cells to mimic highly mutated tumors and showed that cross-presenting cDC1 plays a crucial role in governing the immunogenicity of hypermutated tumors in subcutaneous and orthotopic lung settings. It was observed that simply expressing neoantigens was insufficient to trigger a response to aPD-L1 antibodies. Subsequently, Flt3L-mediated expansion of the cDC compartment effectively amplified cytotoxic CD8+ T cells in high TMB tumors and enhanced reactivity to multiple neoantigens through cDC1. Using scRNA seq of the lung immune infiltrate in tumors with neoantigens, Flt3L/aCD40 therapy prevented tolerogenic programs in lung-resident cDCs. It facilitated the acquisition of an effector CD8+ T cell signature that counteracted exhaustion. Overall, the study highlights the required role of enhancing cDC1 activity for therapeutic efficacy in hypermutated lung tumors that are unresponsive to immune checkpoint blockade (ICB).

Major Concerns:

1. In the lung, it is important to evaluate the separation of intravascular and extravascular cells in tumor samples because one can turn a lung white and still have 70% of its leukocytes in the vasculature. All perfusion does is push out red blood cells. To address this, the

authors should administer intravascularly anti-CD45 antibodies before harvesting to obtain a more precise T-cell assessment.

2. In the s.c. tumors for the tri-therapy approach, there are a number of controls lacking, such as FLT3L+pIC or pIC alone. It is surprising FLT3L and pIC do not affect the KPcont tumors at all.

3. Why did the authors switch therapy from FLT3L/pIC/aPDL1 to FLT3L/aCD40 in the lung?

4. If aCD40 is being used as a combination therapy, then why not examine the CD4 T cell population in this setting?

5. It is unclear if the CD45+ cells infiltrating the TME in lung tissue originate from the extravascular population and are not contaminated by intravascular cells. Furthermore, the authors focused exclusively on DC and CD8+ T cell clusters and did not mention the infiltration of other myeloid cells, T cells, and APCs- and these other infiltrations are not insignificant.

6. Although the amplification of CD8+ T cells was effectively illustrated in KPneo skin and lung tumors. To further validate the effector functions, a ratio analysis of CD8+ T cells to Tregs infiltrating the tumor site support the data further.

7. The gene signature for tolerogenic DCs needs to be clarified. The authors did not prove that these DCs and the genes they express promote tolerance. It is well known that these genes go up in mature activated DCs that differentiate effector T cells, so calling them tolerogenic is incorrect.

Minor comments:

Pg 5: Profiling of s.c. tumors by flow cytometry confirmed a significant enhancement of total

infiltrating CD45+ and --- “cells” is missing after CD45+

Reviewer #3 (Remarks to the Author): with expertise in DCs, cancer immunology

In this manuscript, Lopez-Rodriguez et al. sought to examine the role of cDC1 on the immune control of NSCLC with a high TMB. For this, the authors generated a hypermutated variant of the KP lung cancer cell line by genetically ablating Mlh1 through CRISPR technology and thus rendering the cells mismatch repair deficient. These cells (KPneo) show increased TMB and selective expression of various ‘neoantigens’ compared with control KP cells. Implanting the KPneo cells in the flank of mice or intra-venously, the authors show that cDC1 significantly contribute to T cell-mediated tumor control and that DC-targeting therapies, but not PDL1 blockade, enhance tumor control. By performing single cell RNAseq in KPneo lung lesions of mice treated with a DC-targeting therapy, the authors identify treatment-related transcriptional alterations within the CD8 T cell and cDC compartments.

Considerations:

cDC1 have been previously and extensively shown to critically contribute to T cell-mediated control in numerous experimental settings and models. That they also do it in the context of a single tumor model with increased neoantigen burden is not surprising and particularly noteworthy. This is also the case for the variable DC-targeting therapies tested in the manuscript that include FLT3L and other myeloid cell stimulating compounds. In this regard, despite their claim, the authors do not show that “FLT3L-therapy is necessary and sufficient to achieve therapeutic benefit...”

Major comments:

- To better illustrate a key and potentially specific contribution of cDC1 to T cell-mediated control of TMB high tumors the authors need to consistently compare the response of the KPneo side-by-side with that of the KP control cells. This is essential in multiple experiments including following depletion of CD8 T cells, when using Batf3 KO or XCR1-DTA mice.

- To increase the conceptual advance of the manuscript it would be of interest to also study the role of the 22 neoantigens (or some of them) shared between the KPneo and control cells. This would help substantiate the authors conclusion that the non-shared neoantigens account for the increase control of KPneo cells and that cDC1 contribute to broadening the response to multiple neoAgs.

The authors need to analyze the response of non-tumor-draining LNs alongside tumor-draining LNs across the different readouts.

- Much, if not all, of the immune cell analysis in the different experimental groups is carried out comparing KP tumors of different sizes which makes the interpretation of the data difficult. Authors need to discuss this caveat and show both frequency and number of cells consistently throughout the manuscript.

- In Fig. 2H, a trend towards tumor shrinkage can be observed at the last time point shown for the KPctrl following Flt3/poly I:C/aPD-L1 treatment. A full tumor growth profile extending beyond day 21 is required to show that KPctrl tumors are indeed refractory to the combination treatment.

- The human TCGA data analysis offers limited advance relatively to previous literature. It has been previously shown in multiple studies that CD8 T cells and cDC1 positively correlate within tumors (e.g. Barry et al Nat Med 2018, Botcher et al Cell 2018). In this analysis, I fail to understand based on which data the authors conclude that cDC1 correlate with survival particularly in high TMB tumors. In Figure 4E, the median survival for cDC1 top tumors is very similar for TMB-low and TMB-high groups arguing against this conclusion.

- The authors refer to their treatment for boosting DCs as “Flt3L mediated therapy”. However, their therapeutic regime comprises a combination of Flt3L together with poly I:C and aPDL1 or with aCD40. Each component can have multiple effects in a variety of immune cell populations and thus it is not possible to establish if the efficacy of the treatments relies on the FLT3L (single Flt3L treatment controls would be required) and on the direct effect on cDC1

- The scRNAseq analysis provides a useful resource and it is hypothesis generating. However, the authors do not functionally validate any of the therapy-induced changes described in cDCs.

- For most of the immunofluorescence experiments the authors show only one representative image. Quantification of multiple microscopic fields in independent

tumors/mice per group should be carried out and included.

Minor comments:

- In Figure 3A, data shown correspond to KPneo tumours implanted in wild-type and transgenic animals treated with the DC-therapy combination treatment. The figure legend refers to an isotype control group not shown in the figure.

Reviewer #4 (Remarks to the Author): with expertise in scRNAseq/omics, immunology

In the manuscript entitled “DC targeted therapy breaks immune tolerance against neoantigen in refractory lung adenocarcinoma” Lopez-Rodriguez et al used single-cell RNA sequencing to identify the mechanisms of Flt3L therapy in hypermutated cancers. My comments are restricted to the content and technical aspects of the transcriptomic analysis.

1) It would be beneficial if the authors include a panel depicting the genes used to identify the clusters from fig S4A.

2) The authors write: “Next, we computationally re-clustered T lymphocytes in 10 transcriptionally defined clusters corresponding to CD4 (C0,C2), Tregs (C9), CD8 (C1,3,5,6), $\gamma\delta$ -T cells (C7) and MAIT T cells (C8) (Figure S4B,C). C4 was excluded based on the expression of NKT cell markers.” Could you include in the supplementary or in the text which are these markers used for excluding NKT cells?

3) The authors wrote: “Analysis of sample composition revealed a substantial enrichment in C6 and a moderate increase in C5 upon therapy (Figure 6B).” It is important to perform some statistical assessment of the cell abundancies per group. Currently there are many methods available for such task. Particularly I would recommend miloR (Dann, E., Henderson, N.C., Teichmann, S.A. et al. Differential abundance testing on single-cell data using k-nearest neighbor graphs. Nat Biotechnol 40, 245–253 (2022).

<https://doi.org/10.1038/s41587-021-01033-z>), but any package that performs statistical cell type abundance analysis should work.

4) “Differential gene expression (DEG) uncovered upregulation of cell cycle genes (Stmn1, Ubec2, Birc5, H2-Q10, Cdca8, Cks1b, Cenpv) across the 3 non-naïve CD8+ T cell clusters, particularly in C5 (Figure 6C)” Could the authors change the display of genes in Figure 6A to

separate the dotplots by treatment group? Considering proliferation signatures are present in figure 6A, but they are claimed to be up upon treatment, it would be interesting to already display such division in panel 6A.

5) In figure 6D it is not clear how these scores were calculated. Can you explain that? “ D) Violin plots showing the mean expression of cytotoxic T cell markers (Gzma, Gzmb, Prf1, Tnf, Ifng, Cxcr6) or exhaustion markers (Pdcd1, Ctla4, Tigit, Lag3, Havcr2, Tox, Nrp1) in the indicated clusters in FL/aCD40 vs. control.”

6) Figure 7A: can you replace the cluster numbers by their names? Or include the names together with the numbers? This helps the reader.

7) For cluster abundancy please see comment number 3.

8) For figure 7F, please see comment 5.

Reviewer #1 (Remarks to the Author): with expertise in DCs, cancer immunology

The manuscript by Lopez-Rodriguez et al investigates the immune activation triggered by one NSCLC cancer model with high TMB (KPneo, which was previously described) in mice and the efficacy of immunotherapies in controlling the grafting and outgrowth of those tumors. The authors uncover that KPneo tumors induce a notable (CD8+) T cell response compared with the parental cancer cells when grafted s.c. or orthotopically. Furthermore, KPneo tumors respond to therapies targeting/expanding DCs, but not anti-PD-L1 therapy (at least when grafted s.c.), and anti-CD40+Flt3L treatment causes a remodeling of CD8+ T cells and DCs present in the lung of KPneo tumor-bearing mice. Finally, cDC1 signatures correlate stronger with effector CD8+ T cell presence and fraction survival in TMB-high LUAD patients.

Overall, despite being based on one sole mouse tumor model, this study suggests interesting and novel findings that might be of high value for the community (which is highly appreciated): that TMB-high NSCLC tumors might be refractory to immune checkpoint blockade, but respond to DC expanding/targeting-therapies that remodel DC populations in the tumor-bearing lung. Unfortunately, both findings are poorly experimentally supported or explored, important controls and experimental groups are missing in many experiments in the manuscript, and the study is entirely biased on CD8+ T cells as DC-induced effector cells and CD4+ T cells are not considered (see detailed comments below). If those concerns are addressed, the study will represent a relevant work to be published in Nature Communications.

Major comment 1 – The responses of NSCLC with or without TMB to immune checkpoint therapy and DC-expanding/targeted therapy is poorly addressed

In my view, the differential response to those immunotherapies and a potential dependence on high-TMB probably represents the most novel and relevant finding of this study and, therefore, should be properly and systematically investigated:

In subcutaneous growth experiments (Figure 1-3), treatment with anti-PD-L1 is never compared with polyI:C+Flt3L or aCD40+Flt3L. And, as mentioned in detail below, important controls/experimental groups are missing. To better establish the responsiveness of KPctrl and KPneo tumors to therapies, at least 2 types of immune checkpoint therapies should be tested in both models. This is especially important because anti-PD-L1 is actually (mainly) targeting

cDC1s (PD-L1 is expressed by cDC1s, upregulated upon Ag uptake; and crucial for therapy efficacy, PMID: 35122038 and 32973173). Hence, testing of immune checkpoint antibodies not targeting DCs would be highly relevant, such as anti-Ctla4 or anti-PD1. While it is greatly appreciated that 2 combinations of cDC1-targeting approaches are used in the manuscript, it is entirely unclear why the authors use those in different settings. Please analyze 2 different immune checkpoint therapies and 2 different cDC1-targeting strategies in the same system (ideally in orthotopically grafted tumors) to determine if KPctrl and KPneo tumors respond to immune checkpoint blockade or DC-expanding/targeting therapy (as monotherapies and combination-treatments).

Moreover, the treatment regimen used in Figure 5 is suboptimal and largely precludes seeing effects of immune checkpoint/anti-PD-L1 therapy – which usually takes more than 3 days to have an effect – while Flt3L treatment was given for 8 days and started even before tumor injection. Hence, please administer both treatments at the same time during tumor growth (as for example done in Figure 2F) to be able fairly evaluate the effect of the different therapies (and if they work only in high-TMB tumors or not).

It is intriguing to see that CD8+ T cells in s.c. KPneo tumors and lungs harboring KPneo tumors upregulate PD-1 (at baseline or upon DC-targeted therapy). Yet, anti-PD-L1 blockade is not effective and DC-targeted therapy does not appear to have an additive effect with anti-PD-L1 therapy. Could the authors comment on this surprising finding?

Finally, the first paragraph of the results is entitled “Immunogenic KP encoding bona fide neoAgs are refractory to checkpoint inhibition”. I agree that an entire paragraph should be dedicated to those analyses (see my comments above), however as of now immune checkpoint inhibition was not explored at all in the following paragraphs.

We thank the reviewer for the accurate revision and for recognizing the potential value of our findings. We just wish to kindly disagree on the fact the KP^{neo} was previously shown. Indeed this specific model was generated and characterized by us (including identification of neoantigens) for this study and we believe it may be of value for the community. We acknowledge the confusion created by the way experimental groups and therapies were presented in the original version. We have now performed several new experiments to harmonize therapies and groups and we added the requested controls. The revised version is substantially

improved, revolving on the role of DC and DC-therapy in controlling tumors of diverse neoantigen densities/immunogenicity. Since previous studies firmly established that KP is refractory to PD-1, PD-L1 and PD-1+CTLA4 (Horton et al., 2021; Maier et al., 2020; Martinez-Usatorre et al., 2021; results page 7), we have not further investigated checkpoint inhibitors. We performed one single new experiment to confirm that KP^{ctrl} and KP^{neo} are refractory to anti PD-L1 (revised Figure 2A). The discrepancy with anti PD-L1 efficacy found in other tumor models is discussed at page 15.

The rest of the experiments were performed using a single DC-therapy formulation (FL/ α CD40), harmonized across groups, comparing side-by-side KP^{cont} and KP^{neo} in s.c. and orthotopic models, wild type and DC1 deficient animals. The text, titles of sections and conclusions were modified accordingly, addressing all the concerns expressed in this first comment.

In summary, the new results presented in revised Figure 2 and 5 show that:

- 1) DC therapy given as single therapy strongly inhibits the growth of highly immunogenic tumors in s.c. setting and reduces their progression in orthotopic settings.
- 2) Poorly immunogenic tumors respond partially to DC therapy in s.c. setting and are refractory in the immunosuppressive lung environment.
- 3) Recruitment of CD8 T cells in tumors and specific responses to neoAgs are consistent with the therapeutic efficacy and demonstrate that tumor control is mediated by a DC1-CD8 axis, which is more effective in a neoAgs rich context.
- 4) An extended analysis of the specificity of CD8 T cell responses, including shared neoAgs, unveiled that: 1) less immunogenic epitopes in KP^{cont} become immunogenic in the KP^{neo} context (Revised Fig1E-G) ; 2) DC1 are required to promote broad responses to both shared and unique neoAgs (Figure 1H); 3) The efficacy of DC-therapy in KP^{neo} depends on DC1.

Major comment 2 – It is not clear if the remodeling of T cells and DCs in KP^{neo} tumor-harboring lungs depends on the high TMB burden of those tumors or the presence of tumors at all. Could that be a general feature of α CD40/Flt3L (DC-expanding/targeting) therapy?

The second most relevant finding is the notable remodeling of DC populations (mainly cDC1s) and (likely subsequently?) CD8+ T cell populations upon α CD40+Flt3L therapy. The important remaining question is if this remodeling is unique to NSCLC with high-TMB (or tumor-presence in the lung)? Hence, could the authors investigate the most notable remodeling features (as determined in Figure 6 and 7) in cDC1s, cDC2s and CD8+ T cells (by flow cytometry and/or RT-qPCR) in naïve and KP^{ctrl} tumors-bearing mice (in the latter also the disappearance of mregDCs) treated or not with α CD40+Flt3L therapy? This is important information to better understand

the mechanisms behind the effect of this therapy (is it dependent on the presence of a cancer in the lung? Is the TMB of the cancer relevant?) and, hence, the selection of patients with cancer for this type of therapy.

This is an important aspect that we have addressed experimentally. We have now validated scRNA data by flow cytometry in both KP^{cont} and KP^{neo}.

We found that:

1) Expansion of DCs post-therapy does not depend on the genotype and it is equal in KP^{ctrl} and KP^{neo} in both s.c. and lung settings. The increment in cell numbers, maturation markers and proliferation of DCs are presented in revised Figure S2A-B and revised Figure 7D-H. The disappearance of mReg (defined as PD-L1⁺/CCR7⁺, see below) is observed in the two genotypes. This result is reported in Supplementary Figure 6D, for the KP^{neo} condition.

2) Remodeling of CD8 T cells by therapy, instead, is proportional to neoAgs content. CD8 T cells proliferation, interferon-gamma production, fraction of TCF1⁺PD-1⁺ precursors and downregulation of exhaustion markers are more evident in KP^{neo}, reflecting a larger repertoire of neoAg specific T cells that can be boosted by DCs expansion (Revised Figure 6E-H).

We conclude that DC remodeling occurs independently of the tumor genotype, however, the impact it has on the T cell compartment is proportional to the TMB.

Major comment 3 – Vital controls are missing in tumor growth experiments with/without therapy, the enhanced immune activation in KP^{neo} tumors at baseline is largely omitted and tumor development on cDC1-deficient mice has to be analyzed in the absence of therapeutic treatments and of KP^{ctrl} cells.

The title of the manuscript reads “DC targeted therapy breaks immune tolerance against neoantigen in refractory lung adenocarcinoma”. However, the authors report a significant immune activation in KPneo tumors compared with KPctrl tumors in mice (Fig. 1, 2, S1 and S3) and those cancers progress much slower – so there is clearly no “immune tolerance”, but an immune induction in those high TMB KPneo cancers. Therefore, the title has to be changed (for the combination/comparison with immune checkpoint blockade, please see my major comment1). Moreover, the treatment with polyI:C+Flt3L+anti-PD-L1 or α CD40/Flt3L simply appears to amplify the pre-existing immune induction in KPneo tumors and does not appear to induce additional immune responses. However, the results in Figure 2I+J in the IgG-treated groups are not consistent with the findings of the authors in Figure S1, where the authors already show a significant T cell-mediated immune response in mice grafted with KP-neo tumors compared with KPctrl tumors. Could the authors please explain those discrepancies?

We agree with the wrong usage of “DC therapy breaks immune tolerance” in the title. The new title better reflects the findings “DCs targeted therapy expands CD8 T cell responses to bona-fide neoantigens in lung tumors”.

As explained in reply to comment 1, Figure 2 has been deeply revised and presents the results of α PD-L1 and DC-therapy, separately. The new experiments (revised Figure 2 and S1) confirm findings in the original version, i.e., there is a basal response to KP^{neo} that delays tumor growth. Upon DC-therapy, the responses to KP^{neo} are amplified achieving rejection of a large fraction of tumors. Conversely, KP^{ctrl} tumors that do not induce basal responses are mildly warmed up by therapy, recruitment/retention of CD8 in tumor tissue remains low and tumor progression is delayed but not blocked.

It has to be noted that the combination FL+polyI:C + α PD-L1 presented in the original version, had no effect on progression of KP^{ctrl} and only partially reduced growth of KP^{neo}. The stronger impact of DC-therapy given as single axis (w/o α PD-L1) in the new experiments may be due to the different adjuvants (α CD40 instead of polyI:C) or the different administration routes (systemically as opposed to intranodal, see below).

In the s.c. grafted tumors, while it is clear that anti-PD-L1 alone does not reduce the growth of KPneo tumors, this manuscript lacks the essential control of treatment of KPctrl and KPneo tumors with polyI:C+Flt3L only (without immune checkpoint inhibition). Is this treatment causing the tumor growth difference and what is the relevance of anti-PD-L1 treatment in this setting? Also, could the authors please quantify cDC1 presence and activation (e.g. activation markers CD40, CD80, CD86) in s.c. tumors and tdLNs of KPctrl and KPneo-tumour bearing mice being untreated, treated with anti-PD-L1 monotherapy and polyI:C+Flt3L therapy (are cDC1s equally enriched and

activated in KPctrl as KPneo-bearing mice and how do the treatments affect this?), especially because the authors chose an unusual intratumoral administration route for Flt3L and there is a massive size difference in examined tumors. cDC1 presence is only quantified upon aCD40/Flt3L treatment in the lung of KPneo tumor-bearing mice, however it remains unclear if anti-PD-L1 affect cDC1 presence or activation (given that those cells express this co-inhibitory molecule and might be depleted?)? Also, could the authors use flow cytometry to quantify cDC1s and cDC2s (as it was done in Figure S5) in lungs and s.c. grafted tumors, as this is much more quantitative.

As stated in the previous reply, we have now administered DC-therapy systemically (the new formulation Flt3L+ α CD40), uncoupled from anti-PD-L1. The results indicate that DC-therapy is having a major impact (proportional to neoAgs levels) whereas anti-PD-L1 has not.

We now quantified by flow cytometry the presence and activation of DC1 in the tumor draining LNs and contralateral LNs, at the time when therapy was discontinued and tumors had similar sizes. Results presented in revised Figure S2 A-B, show that therapy expands and mature DC1, independently of the tumor genotype and systemically (see also the above comment for orthotopic model). The tumor mass at this time point was too small to quantify DCs.

Furthermore, more details are needed for the choice of route of immunotherapy administration. Firstly, could the authors please explain how anti-CD40, Flt3L and anti-PD-L1 was administered to mice with orthotopic KPneo tumors in the lung (I couldn't find this information in the main text, Figure legends or the Methods) – intratumorally as well? Secondly, why was the intratumoral route chosen for polyI:C, Flt3L and anti-PD-L1 (other than Salmon et al administering polyI:C intratumorally)? Intratumoral doesn't appear to be a feasible administration route for patients with lung cancer. Moreover, is not used for Flt3L or anti-PD-L1, which are usually given intraperitoneally (for example, in the study Salmon et al, Flt3L is administered intra-peritoneally, because it has to reach the bone marrow rather than the tumor to expand cDC1s).

Could the administration of anti-PD-L1 intratumorally instead of interperitoneally account for the non-responsiveness of s.c. grafted KPneo tumors? PD-L1 in lung KP^{neo} tumors.

Please comment on those issues and, importantly, clarify the administration route for the tumors grafted in the lung. Ideally, for the experiments comparing immunotherapies mentioned in major comment 1, a feasible and commonly used administration route should be chosen – additional to a treatment regimen that allows for side-by-side comparison of those immunotherapies.

In the new experiments presented in Revised Figure 2, we have administered Flt3L+aCD40 systemically like in the orthotopic model to solve this inconsistency. The missing information is added methods at page 21.

We have repeated experiments with systemic administration of anti-PD-L1 in the s.c. model. In line with previous reports by other groups, we have seen no effect of the therapy in KP^{ctrl} or KP^{neo}.

Why were the experiments using Batf3-ko and XCR1-DTA separated in the manuscript? That causes confusion. Moreover, could the authors please report the tumor growth and T cell activation of KP^{ctrl} cells grafted onto those cDC1-deficient mouse models, with or without therapy? This is important to understand if the observed effects are due to the altered TMB or simply due to cDC1 loss (and hence occurring in the parental cancer as well). Also, how is tumor growth of KP^{neo} cells on Batf3-ko and XCR1-DTA mice without therapy – is cDC1 depletion causing the effect or does it only happen upon therapy? The dramatic reduction of CD8⁺ T cells in untreated Batf3-ko mice grafted with KP^{neo} tumors suggests a therapy-independent effect of cDC1 loss. Moreover, to understand the mechanism, could the authors investigate the effects of polyI:C+Flt3L independently of anti-PD-L1 on those cDC1-deficient mouse models? Those points have to be addressed before the authors can conclude that “cDC1 (...) contribute to expand the intensity and diversity of CD8⁺ T cell responses to neoAgs”

We fully agree with these comments and we have performed all the suggested experiments. Figure 1C and S1G show the growth curve of KP^{ctrl} and KP^{neo} and CD8 recruitment in wt and Batf3^{-/-} recipients without treatment. Results indicate that spontaneous responses to KP^{neo} are, at least in part, mediated by DC1. Growth of KP^{ctrl} is equal in wt and Batf3^{-/-}, in agreement with poor basal immunogenicity and lack of effect of CD8 depletion.

In the context of DC-therapy (only FL/aCD40 w/o PD-L1) (Figure 3A-C), rejection of KP^{neo} is totally abrogated in Batf3^{-/-}, indicating a major impact of DC1 in mediating therapy-driven anti-tumoral effects in neoAgs rich tumors.

The milder effect that DC-therapy has in KP^{ctrl}, is only partially reduced in Batf3^{-/-} mice, suggesting DC-independent axis that may prevail in low antigenic tumors. We conclude that both spontaneous and therapy-induced responses to KP^{neo} depend on Batf3 DC1.

The new experiments were performed using a single model of DC1 depletion (Batf3^{-/-}) as the XCR1-DTA strain was not available from collaborators during the time frame of this revision.

Finally, were tumors analyzed at a timepoint when they had a similar tumor size or at endpoint (please see rationales for that in my comments below). If

not, please analyze equal-size tumors wherever possible and, otherwise, clearly state the caveat of having to investigate differently sized tumors for every experiment.

In the new experiments, we performed analysis of the T cell infiltrate at day 12 when therapy was discontinued and tumors had similar sizes. The analysis was repeated at the endpoint, showing similar trends (Revised 2 D-G). Overall the data confirm (see above) that therapy amplifies the recruitment/retention of CD8 T cell in tumor tissues, which is most prominent for KP^{neo} tumors.

Major comment 4 – The authors unfortunately solely focus on CD8+ T cells and ignore the emerging role of CD4+ Th1 cells in anti-cancer immunity and the importance of cDC1s controlling their activation and vice versa (PMID: 32457487; 32788723 and 30961656).

The connection between cDC1s and CD4+ (Th1) T cells is strongly emerging in the literature, which should be considered by the authors in this study. Considering the importance of anti-tumor functions of CD4+ Th1 T cells in cancer and the potency of cDC1s to induce those cells (and their cross-talk; PMID: 32457487; 32788723 and 30961656), the induction of CD4+ T cell responses and their relevance for growth of KP-Neo cancers should also be determined to better understand the experimental high TMB model that the authors use and the underlying mechanism.

Firstly, could the CD4+ T cell clusters (as shown in Figure S4B) be analyzed in-depth, as was done for CD8+ T cells in Figure 6. Does DC-expanding/targeting therapy also remodel CD4+ T cells, or is that exclusive to CD8+ T cells?

Secondly, in Figure 1-3, the authors did not formally show that it is CD8+ T cells that are specific for the neoantigens expressed by the KP-Neo cells (for example, total IFN γ was measured after re-stimulation of the entire LN). Also, according to the nature portfolio reporting summary, CD8+ T cells were gated as being CD45+ CD3+ (This information is entirely missing from the methods and should be added). Hence, could the authors re-analyze their data and focus on CD3+ CD8-negative cells (which are largely CD4+ T cells) in their respective Flow Cytometry analyses in Figure 1, 2, 3, 5D, S1, S2 and S3? Especially the number/frequency of those CD3+ CD8- T cells, their expression of CD62L, CD44, PD-1 and IFN γ in the different tumors and treatment regimens?

The addition of those analyses of CD8- T cells and the CD4+ T cell clusters of the scRNAseq will add vital information on the activities of cDC1s controlling all cancer-relevant T cell subsets in cancer (especially since cDC1s are known

to induce/crosstalk with CD4+ (Th1) cells in cancer; PMID: 32457487; 32788723 and 30961656) in a TMB-high setting and will only require re-analysis of already collected data.

The focus of this study is cross-presentation of neoAgs by DC1 and the impact on CD8 T cell activation. While we fully contemplate the role of class-II neoantigens in KP^{neo} and the contribution of CD4 T cells to anti-tumoral responses, this aspect will be extensively addressed in future studies (see discussion page 24 middle paragraph). We have nevertheless added a preliminary analysis of CD4 in the revised version. Data presented in Figure S2 D-E report CD4 proportions and absolute numbers before and after therapy in the s.c. model and Figure S3B in the orthotopic model. In both models, we detected a basally higher number of CD4 T cells infiltrating KP^{neo} tumors, as compared to KP^{ctrl}, in untreated mice. However, CD4 were not significantly enhanced by therapy. In addition, we are now presenting an analysis of the two CD4 clusters (C0,C2) in Supplementary Datafile 4. These tables include DEGs and biological processes from scRNA seq and are commented in the text (results page 12 first paragraph). Concerning the specificity of CD8 T cells responses, all flow data are performed gating on CD3⁺CD8⁺ T cells (CD44, IFN- γ , PD-1). We carefully revised text, legend and methods to make sure this is clear to readers. Moreover, all the new data from ELISpot presented in the revised version are performed using CD8 T cells purified from the spleen.

Finally, please state in the legend of Figure 1E at which days anti-CD8 was administered. Depleting CD8⁺ cells only 3 days after tumor cell injection might aide tumor engraftment rather than the out-growth of cancers (and this information should be easy-access for the reader). Hence, is tumor progression of KP-Neo tumors also accelerated when CD8⁺ T cells are depleted after 12 days, when the KP-ctrl tumors start to outgrow KP-neo tumors? Also, does CD8⁺ T cell depletion affect growth of KP-Neo tumors specifically, or in other words, how is the growth of KPctrl tumors affected after CD8⁺ T cell depletion?

We now show in revised Figure 1A, the growth curve of both KP^{ctrl} and KP^{neo} in the presence of depleting antibodies. Anti-CD8 depleting antibodies were administered at day 3,6,9,12 and 15. We apologise for the missing information that is now present in methods.

The authors write: “Whole exome sequencing showed an increase in single nucleotide variant (SNVs) and frameshifts as compared to KP control cells (KPctrl), which corresponded to 66 and 126 predicted MHC class-I epitopes, respectively (Figure S1B).” Unfortunately, it is not well explained how the number of predicted MHC-I epitopes (and binding affinity) was determined.

The methodology for the prediction of MHC class-I epitopes is well-established. The details and the bioinformatic tools for the prediction are illustrated in the methods section.

Please add some details to the Figure legend.

How did the authors control that KPKO2 (KPneo) cells did not accumulate further detrimental mutations affecting tumor progression (after the analysis in Figure 1) due to their deficiency in Mlh1? Especially when growing in vivo?

We are aware of this problem. To mitigate this, we have initially frozen large stock of cells and always kept passages to a minimum. Nevertheless, we cannot exclude that more mutations accumulate over passages and in vivo. Still, our data demonstrate that reactivity to encoded neoAgs identified early on was maintained for the period of these experiments (more than 2 years), suggesting that clonal antigens are preserved.

How and why was interference with Mlh1 chosen – is it frequently mutated in NSCLC and correlative with high TMB? Please explain the rationales for this experimental approach to induce a high TMB.

Mlh1 loss occurs in NSCLC, although it is not the most frequent mutation underlying high TMB in this tissue (PMID: 28596308). We have chosen it to enhance the TMB as this was previously shown to be a reliable method to mimic mutated lung tumors (PMID: 34380768).

Figure S1F-J should be repeated at day 12 of tumor growth – when ctrl and neo tumors have an equal size. Such an analysis is much more meaningful, because the results of the analysis performed by the authors at end stage likely reflect the difference in tumor size/progression and it is unclear if that is the cause for differential tumor growth. Only an analysis of tumors at a time before the growth difference between ctrl and neo becomes apparent will answer this question. In line, at which time point was the analysis in Figure 1F performed? It should be done (with at least a few neoantigens) at around day 12 as well. Same for Figure 2A+B, please analyze cDC1 presence in tumors at day 12 and not when the microenvironments are so dramatically different due to tumor size. Alternatively, please clearly state in the text that tumors were analyzed as such different sizes and the caveats that is poses on the interpretation of the results.

As commented above we have performed the analysis and ELISpot at day 12 (Revised Figure 1,2,3).

Title of Figure 7: it should read “Flt3L”. And, how do the authors know that those cDC1s are lung-resident and have not been recently recruited

(especially pre-cDC1s) by the tumors? Or do the authors assume it is a specific activation of lung pre-cDC1s (as the cDC1b cluster is proliferating)?

As anticipated in reply to main comment 2, we have now validated the remodelling of the DC compartment by flow using XCR1-venus reporters mice and Ki67 labeling (revised Figure 7). We confirm that a fraction of lung cDC1 with low XCR1 expression (cDC1b) is proliferating and may represent pre-DC1. Based on the current approach, we cannot discriminate whether lung DC1 boosted by therapy have been recruited from the circulation or have proliferated in lung tissues. We will rephrase the conclusions accordingly.

Figure S5D – could the authors provide a complete list of deregulated genes as a supplementary excel file?

In Supplementary DataFile 3 and 6 we provide a list of DEGs and GSEA for T and DC clusters.

The authors write “Altogether, expansion of pre-cDC1 and cDC1 (...) replenishes the lung with naive cDCs with increased ability to sample and present tumor antigens”. However, the authors did not actually test the ability of cDC1s for phagocytosis/AG sampling or their AG processing or presentation capacity. Related genes were also not mentioned to be deregulated in the scRNAseq analysis. Hence, could the authors please rephrase their statement or assess those functional traits of cDC1s?

We have now added key experiments in Revised Figure 7, to validate by flow some changes observed by scRNA, such as proliferation and reduction of markers of exhaustion (PD-L1 and Tim3), post-therapy. Moreover, we have performed a functional assay using DC1 sorted from therapy-treated or control lungs to assess the capacity to prime CD8 T cells. Results presented in Figure 7I-K show an improved capacity to activate CD8 T cells.

Font in Figures is generally quite small and hard to read.

Reviewer #2 (Remarks to the Author): with expertise in lung immunology/DC, cancer immunology

Rodriguez et al. generated a hypermutated variant of KP lung adenocarcinoma cells to mimic highly mutated tumors and showed that cross-presenting cDC1 plays a crucial role in governing the immunogenicity of hypermutated tumors in subcutaneous and orthotopic lung settings. It was observed that simply expressing neoantigens was insufficient to trigger a response to aPD-L1

antibodies. Subsequently, Flt3L-mediated expansion of the cDC compartment effectively amplified cytotoxic CD8+ T cells in high TMB tumors and enhanced reactivity to multiple neoantigens through cDC1. Using scRNA seq of the lung immune infiltrate in tumors with neoantigens, Flt3L/ α CD40 therapy prevented tolerogenic programs in lung-resident cDCs. It facilitated the acquisition of an effector CD8+ T cell signature that counteracted exhaustion. Overall, the study highlights the required role of enhancing cDC1 activity for therapeutic efficacy in hypermutated lung tumors that are unresponsive to immune checkpoint blockade (ICB).

Major Concerns:

1. In the lung, it is important to evaluate the separation of intravascular and extravascular cells in tumor samples because one can turn a lung white and still have 70% of its leukocytes in the vasculature. All perfusion does is push out red blood cells. To address this, the authors should administer intravascularly anti-CD45 antibodies before harvesting to obtain a more precise T-cell assessment.

We have now repeated key T cell readouts (Ki67, Granzyme and markers of exhaustion) upon exclusion of circulating cells by intravenous delivery of a pan-leukocytes antibody. Results are displayed in Figure S5A-D and show that tissue-resident CD8 T cells are responding upon therapy. This is also corroborated by IHC data showing infiltration of tumor nodules.

2. In the s.c. tumors for the tri-therapy approach, there are a number of controls lacking, such as FLT3L+pIC or pIC alone. It is surprising FLT3L and pIC do not affect the KP^{cont} tumors at all.

3. Why did the authors switch therapy from FLT3L/pIC/ α PDL1 to FLT3L/ α CD40 in the lung?

We agree with the reviewer that the data presented in the original version were not coherent and generated confusion.

We have now generated a substantial amount of new data focusing on DC therapy. We administered a single DC-therapy formulation (FL/ α CD40, systemic), harmonized across groups, comparing side-by-side KP^{cont} and KP^{neo} in s.c. and orthotopic models, wild type and DC1 deficient animals. Since previous studies firmly established that KP is refractory to PD-1, PD-L1 and PD-1+CTLA4 (Horton, Maier, Sanchez Paulete, de Palma), we have not further investigated checkpoint inhibitors. We performed one single new experiment to confirm that KP^{ctrl} and KP^{neo} are refractory to anti PD-L1 (revised Figure 2A). Given that Flt3L was

previously shown to require DC activation (PMID: 27096321, PMID: 32183949), we have not included groups to test Flt3L and α CD40, separately.

The new results are presented in revised Figure 2 and show that FL/ α CD40 reject a large fraction of KP^{neo} and has a partial impact on the growth of KP^{ctrl}, showing an overall better efficacy (proportional to neoAgs content) than the previous combination (Flt3L+poli:IC+aPD-L1). We speculate this may be due to 1) the different administration routes (intranodal vs systemic injection) or 2) the possible toxic effects of intratumoral poli:IC.

4. If α CD40 is being used as a combination therapy, then why not examine the CD4 T cell population in this setting?

The focus of this study is cross-presentation of neoAgs by DC1 and the impact on CD8 T cell activation. While we fully contemplate the role of class-II neoantigens in KP^{neo} and the contribution of CD4 T cells to anti-tumoral responses, this aspect will be extensively addressed in future studies. We have nevertheless added a preliminary analysis of CD4 in the revised version. Data presented in Figure S2 D-E report CD4 proportions and absolute numbers before and after therapy in the s.c. model and Figure S3B in the orthotopic model. In both models, we detected a basally higher number of CD4 T cells infiltrating KP^{neo} tumors, as compared to KP^{ctrl}, in untreated mice. However, CD4 were not significantly enhanced by therapy. In addition, we are now presenting an analysis of the two CD4 clusters (C0,C2) in Supplementary Datafile 4. These tables include DEGs and biological processes from scRNA seq and are commented in the text (results page 12).

5. It is unclear if the CD45+ cells infiltrating the TME in lung tissue originate from the extravascular population and are not contaminated by intravascular cells. Furthermore, the authors focused exclusively on DC and CD8+ T cell clusters and did not mention the infiltration of other myeloid cells, T cells, and APCs- and these other infiltrations are not insignificant.

We followed the same procedure of a seminal study describing the immune cell structure of KP lung tumors and its analogy with human lung tumor (PMID: 30979687) (perfusion and isolation of CD45), obtaining comparable results. We have now added a validation of scRNAseq data by flow cytometry upon the exclusion of circulating cells by intravenous delivery of a pan-leukocytes antibody (Figure S5). The data show that extravascular cells show the same changes observed when collecting the total fraction, which indicates that the impact of therapy is evident on both intravascular and extravascular cells.

Since data in Figure 3 show that therapy efficacy is lost in Batf3^{-/-} null mice, we have focused the analysis on DC1 and activation of CD8 T cells by cross-presentation. The effect of DC-therapy on the other clusters was beyond the present scope and it will be analyzed in future studies.

6. Although the amplification of CD8+ T cells was effectively illustrated in KPneo skin and lung tumors. To further validate the effector functions, a ratio analysis of CD8+ T cells to Tregs infiltrating the tumor site support the data further.

In the new experiments presented in revised Figure 2G we have analyzed the CD8 T cell /T regs ratio. The data show an increment after therapy, in the KP^{neo} group.

7. The gene signature for tolerogenic DCs needs to be clarified. The authors did not prove that these DCs and the genes they express promote tolerance. It is well known that these genes go up in mature activated DCs that differentiate effector T cells, so calling them tolerogenic is incorrect.

We thank the reviewer for asking clarification on this aspect and the terminology. To define the signature we have used some of the genes that were associated to regulatory DCs functions by Maier et al. (PMID: 32269339) and added genes that were associated to DCs inhibitory functions (IL10ra, Aldh1a1, Tgfbr1). The program has been called “exhausted DC program” (violins in Figure 7C), better commented in results (page 19 last paragraph). In revised Figure 7, we provide additional data to further document functional changes in DCs after therapy.

Minor comments:

Pg 5: Profiling of s.c. tumors by flow cytometry confirmed a significant enhancement of total infiltrating CD45+ and --- “cells” is missing after CD45+

Reviewer #3 (Remarks to the Author): with expertise in DCs, cancer immunology

In this manuscript, Lopez-Rodriguez et al. sought to examine the role of cDC1 on the immune control of NSCLC with a high TMB. For this, the authors generated a hypermutated variant of the KP lung cancer cell line by genetically ablating Mlh1 through CRISPR technology and thus rendering the cells mismatch repair deficient. These cells (KPneo) show increased TMB and selective expression of various ‘neoantigens’ compared with control KP cells. Implanting the KPneo cells in the flank of mice or intra-venously, the authors show that cDC1 significantly contribute to T cell-mediated tumor control and that DC-targeting therapies, but not PDL1 blockade, enhance tumor control. By performing single cell RNAseq in KPneo lung lesions of mice treated with a DC-targeting therapy, the authors identify treatment-related transcriptional alterations within the CD8 T cell and cDC compartments.

Considerations:

cDC1 have been previously and extensively shown to critically contribute to T cell-mediated control in numerous experimental settings and models. That they also do it in the context of a single tumor model with increased neoantigen burden is not surprising and particularly noteworthy. This is also the case for the variable DC-targeting therapies tested in the manuscript that include FLT3L and other myeloid cell stimulating compounds. In this regard, despite their claim, the authors do not show that “FLT3L-therapy is necessary and sufficient to achieve therapeutic benefit...”

We would like to underline that despite the well-accepted role of DC1 in antitumor immunity, little has been done at the preclinical level to understand the regulation of responses to complex pattern of bona-fide neoAgs, beyond single artificial antigens, by DCs. Similarly, DC- therapy has been little explored in lung tumors which are refractory to checkpoint inhibitors and remodelling of immune cells induced by therapy has not been dissected.

Major comments:

- To better illustrate a key and potentially specific contribution of cDC1 to T cell-mediated control of TMB high tumors the authors need to consistently compare the response of the KP^{neo} side-by-side with that of the KP control cells. This is essential in multiple experiments including following depletion of CD8 T cells, when using Batf3 KO or XCR1-DTA mice.

We agree with the need to revise the experimental design comparing side-by-side the response of KP^{ctrl} and KP^{neo} in all experiments. We have now performed new experiments using a single DC-therapy formulation (FL/αCD40), harmonized across groups, comparing side-by-side KP^{cont} and KP^{neo} in s.c. and orthotopic models, in wild type and DC1 deficient animals.

In the revised version, we focus on DC-therapy as single axis and we have not repeated the combination with anti-PD-L1, since previous studies already established that KP is refractory to PD-1, PD-L1 and PD-1+CTLA4.

The new results are presented in revised Figure 2,3 and 5 and show that:

- 1) DC therapy given as single therapy strongly inhibits the growth of immunogenic KP^{neo} tumors in s.c. setting and reduces their progression in orthotopic settings.**
- 2) Poorly immunogenic KP^{ctrl} tumors respond partially to DC therapy in s.c. setting and do not respond in the immunosuppressive lung environment.**
- 3) DC1 deletion abrogates spontaneous and therapy-driven responses to KP^{neo} and has a marginal impact on KP^{ctrl}, in line with the overall reduced immunogenicity.**

- To increase the conceptual advance of the manuscript it would be of interest to also study the role of the 22 neoantigens (or some of them) shared between the KPneo and control cells. This would help substantiate the authors conclusion that the non-shared neoantigens account for the increase control of KPneo cells and that cDC1 contribute to broadening the response to multiple neoAgs. The authors need to analyze the response of non-tumor-draining LNs alongside tumor-draining LNs across the different readouts.

This is a very interesting point. We have now extended the analysis to shared peptides in the context of spontaneous responses, therapy-driven responses and Batf3^{-/-} mice. Results are presented in revised Figure 1 (E-H), 2 (H-I) and 3 (G).

The data show that:

1) Animals challenged with KP^{ctrl} do not induce responses to shared antigens but KP^{neo} challenged animals mount responses to unique neoAgs as well as shared neoAgs (see results, page 6-7-8). This result was confirmed for one shared neoAgs by dextramers-based identification of neoAgs specific T cells. This reinforces that unique antigens drive antitumoral T cell responses. Moreover, it shows that less immunogenic shared epitopes can induce responses in the context of more immunogenic tumors. This is highly consistent with a recent study showing that reactivity to suboptimal antigens can be promoted by the presence of stronger immunogenic antigens via modulation of DCs immunostimulatory properties (PMID: 37548358).

- Much, if not all, of the immune cell analysis in the different experimental groups is carried out comparing KP tumors of different sizes which makes the interpretation of the data difficult. Authors need to discuss this caveat and show both frequency and number of cells consistently throughout the manuscript.

This point, also raised by rev 1, has been addressed. In the new experiments comparing the impact of DC-therapy on KP^{ctrl} and KP^{neo}, we have added analysis of the T cell infiltrate at day 12, when therapy was discontinued and tumors had similar sizes. The analysis was repeated at the endpoint, showing similar trends (Revised 2 D-G). Overall the data confirm (see above) that therapy amplifies the recruitment/retention of CD8 T cell in tumor tissues, which is most prominent for KP^{neo} tumors.

- In Fig. 2H, a trend towards tumor shrinkage can be observed at the last time point shown for the KP^{ctrl} following Flt3/poly I:C/aPD-L1 treatment. A full tumor growth profile extending beyond day 21 is required to show that KP^{ctrl} tumors are indeed refractory to the combination treatment.

In the new experiments we have uncoupled DC targeting and anti PD-L1. We show in revised Figure 2A that that both KP^{ctrl} and KP^{neo} are refractory to anti-PD-L1. We show in revised Figure 2B that DC-therapy (FL/aCD40) has a strong impact on

KP^{neo} and a moderate impact on KP^{ctrl} (Figure 2 A-C), performing better than the original combination FL+poly:IC+ α PD-L1. We speculate this may be due to the different administration route (systemic vs intranodal) or the different adjuvant (aCD40 vs poly:IC) or a possible negative impact of anti-PD-L1. We have rephrased conclusions accordingly (page 8 results, first paragraph).

- The human TCGA data analysis offers limited advance relatively to previous literature. It has been previously shown in multiple studies that CD8 T cells and cDC1 positively correlate within tumors (e.g. Barry et al Nat Med 2018, Botcher et al Cell 2018). In this analysis, I fail to understand based on which data the authors conclude that cDC1 correlate with survival particularly in high TMB tumors. In Figure 4E, the median survival for cDC1 top tumors is very similar for TMB-low and TMB-high groups arguing against this conclusion.

The novel angle in this analysis is the stratification across diverse TMB and the finding that the highest CD8 effector scores are in TMB^{high} DC1^{high} group, in line with our mouse data. We agree that the conclusion on survival was overstated as the advantage in the cDC1/TMB top tumor was very minor. We have rephrased the conclusions accordingly.

- The authors refer to their treatment for boosting DCs as “Flt3L mediated therapy”. However, their therapeutic regime comprises a combination of Flt3L together with poly I:C and α PDL1 or with aCD40. Each component can have multiple effects in a variety of immune cell populations and thus it is not possible to establish if the efficacy of the treatments relies on the FLT3L (single Flt3L treatment controls would be required) and on the direct effect on cDC1

We have now administered DC therapy (that includes Flt3L+aCD40) uncoupled from anti PD-L1. As the purpose is to compare responses of tumors at different TMB, rather than different therapies, and given that Flt3L was previously shown to require DC activation (PMID: 27096321, PMID: 32183949), we have not included groups to test Flt3L and α CD40, separately.

- The scRNAseq analysis provides a useful resource and it is hypothesis generating. However, the authors do not functionally validate any of the therapy-induced changes described in cDCs.

We have substantially revised Figure 7 to improve the functional validation of the scRNA data, according to this suggestion.

Using the XCR1-Venus reporter mouse we could discriminate XCR1 high and XCR1 low populations (Revised 7D). The latter (XCR1 low) is highly proliferative and

increases in numbers, suggesting their identity as precursors (cDC1b cluster)(Revised 7E). XCR1 high expand in numbers, are less proliferative and show effector functions (IL-12 production) upon therapy, in line with their identity as differentiated cells (cDC1a)(Revised F). Flow cytometry also showed that therapy decreased the expression of inhibitory receptors such as PD-L1 and TIM3 (Revised 7F,G). Importantly, DC1 isolated from tissues of therapy-treated mice were more efficient than controls to activate T cell ex-vivo (Revised 7I-K).

- For most of the immunofluorescence experiments the authors show only one representative image. Quantification of multiple microscopic fields in independent tumors/mice per group should be carried out and included.

We have quantified the infiltration of tumor nodules by IHC (Revised 5F). The other images are representative micrographs accompanied by Flow cytometry quantifications.

Minor comments:

- In Figure 3A, data shown correspond to KPneo tumours implanted in wild-type and transgenic animals treated with the DC-therapy combination treatment. The figure legend refers to an isotype control group not shown in the figure.

We have corrected the mistake in revised Figure 3.

Reviewer #4 (Remarks to the Author): with expertise in scRNAseq/omics, immunology

In the manuscript entitled “DC targeted therapy breaks immune tolerance against neoantigen in refractory lung adenocarcinoma” Lopez-Rodriguez et al used single-cell RNA sequencing to identify the mechanisms of Flt3L therapy in hypermutated cancers. My comments are restricted to the content and technical aspects of the transcriptomic analysis.

1) It would be beneficial if the authors include a panel depicting the genes used to identify the clusters from fig S4A.

We have now added a dot plot showing the main genes used to identify the clusters(Revised S4B). Supplementary DataFile 1, 2 and 5 contain the full list of marker genes used to identify all immune populations, T cell and DCs, respectively

2) The authors write: “Next, we computationally re-clustered T lymphocytes

in 10 transcriptionally defined clusters corresponding to CD4 (C0,C2), Tregs (C9), CD8 (C1,3,5,6), $\gamma\delta$ -T cells (C7) and MAIT T cells (C8) (Figure S4B,C). C4 was excluded based on the expression of NKT cell markers.” Could you include in the supplementary or in the text which are these markers used for excluding NKT cells?

We now list genes used to exclude NKT cells in the text and provide a file with all genes defining the clusters (Results page 16, Supplementary DataFile2).

3) The authors wrote: “Analysis of sample composition revealed a substantial enrichment in C6 and a moderate increase in C5 upon therapy (Figure 6B).” It is important to perform some statistical assessment of the cell abundancies per group. Currently there are many methods available for such task. Particularly I would recommend miloR (Dann, E., Henderson, N.C., Teichmann, S.A. et al. Differential abundance testing on single-cell data using k-nearest neighbor graphs. Nat Biotechnol 40, 245–253 (2022). <https://doi.org/10.1038/s41587-021-01033-z>), but any package that performs statistical cell type abundance analysis should work.

We cannot apply statistics as single-cell data are single points. They come from a pool of 3 independent mice, preserving biological variability. Authors performing the bioinformatic analysis in this paper have used the same criteria (PMID: 37914939).

4) “Differential gene expression (DEG) uncovered upregulation of cell cycle genes (Stmn1, Ubec2, Birc5, H2-Q10, Cdca8, Cks1b, Cenvp) across the 3 non-naïve CD8+ T cell clusters, particularly in C5 (Figure 6C)” Could the authors change the display of genes in Figure 6A to separate the dotplots by treatment group? Considering proliferation signatures are present in figure 6A, but they are claimed to be up upon treatment, it would be interesting to already display such division in panel 6A.

We cannot show dot plot as pre and post therapy, as genes are differentially expressed in the various clusters and the intercluster differences would hide intracluster variations. We have now added functional validation of the proliferative phenotype pre and post therapy in revised Figure 6.

5) In figure 6D it is not clear how these scores were calculated. Can you explain that? “ D) Violin plots showing the mean expression of cytotoxic T cell markers (Gzma, Gzmb, Prf1, Tnf, Ifng, Cxcr6) or exhaustion markers (Pdcd1, Ctla4, Tigit, Lag3, Havcr2, Tox, Nrp1) in the indicated clusters in FL/aCD40 vs. control.”

The values are not scores but mean expression, legend have been modified accordingly:

Violin plots show the distribution of log-transformed means of normalized expression values of cytotoxic T cell markers (Gzma, Gzmb, Prf1, Tnf, Ifng, Cxcr6) or exhaustion markers (Pdcd1, Ctla4, Tigit, Lag3, Havcr2, Tox, Nrp1) in cells of the indicated clusters in FL/aCD40 vs. control.

6) Figure 7A: can you replace the cluster numbers by their names? Or include the names together with the numbers? This helps the reader.

This has been done.

7) For cluster abundancy please see comment number 3.

8) For figure 7F, please see comment 5.

REVIEWER COMMENTS

Reviewer #1 (Remarks to the Author):

The manuscript Lopez-Rodriguez et al underwent a major revision that importantly changed the focus of the study. This includes notable amounts of new experimental data, but unfortunately also the removal of conceptually most novel/relevant data. Certainly, the authors improved the quality of the designs and related conclusions of selected experimental settings, which is highly appreciated. Unfortunately, the probably most relevant finding of the manuscript (that TMB-high NSCLC tumors are refractory to immune checkpoint blockade, but respond to DC expanding/targeting-therapies that remodel DC populations in the tumor-bearing lung) seems to be even less investigated in the revised compared with the initial manuscript. Instead, the authors focused on the relevance of Flt3L+anti-CD40 therapy for lung cancer with/without high TMB. This change of focus of the manuscript is clearly reflected in the change of the title from “DC targeted therapy breaks immune tolerance against neoantigen in refractory lung adenocarcinoma” to “DCs targeted therapy expands CD8 T cell responses to bona-fide neoantigens in lung tumors”. In light of pre-existing literature (for example PMID: 10415026, 34534439) and a report since first manuscript submission (PMID: 37548358), this new conceptual direction has notably lowered the novelty of the manuscript in my view. I would recommend maintaining the initial message of the manuscript (DC therapy being more effective/enabling immune checkpoint efficacy in otherwise refractory high-neoantigen lung cancers) to reach the relevance required for publication in Nature Communication. This includes the side-by-side comparison of immune checkpoint and DC therapy in lung KP-neo tumors (and ideally the combo therapy) as the authors showed in old Figure 5 (but with comparable aPD-L1 and Flt3L/aCD40 treatment regimens) and was suggested in my Major comment 1 in the course of the first revision.

Major comment 1 – The responses of NSCLC with or without TMB to immune checkpoint therapy and DC-expanding/targeted therapy is poorly addressed. In my view, the differential response to those immunotherapies and a potential dependence on high-TMB probably represents the most novel and relevant finding of this study and, therefore, should be

properly and systematically investigated.

It was very much appreciated that the authors used two independent strategies to deplete cDC1s (Batf3-ko and XCR1-DTA) and enhance DC expansion/efficacy (polyI:C+Flt3L and CD40+Flt3L), despite using only one grafted cancer model in mice. Unfortunately, in the revised manuscript, this was reduced to 1 cancer model, 1 DC-deficiency model and 1 DC expansion strategy. Moreover, those DC-targeted strategies are suboptimal and not specific for cDC1s: the Batf3-ko model also targets CD8+ and regulatory T cells (for example PMID: 29147008, 32989328 and 32669309) and anti-CD40 antibodies as well as Flt3L target/expand several DC subsets and other cells (PMID: 33810248 and 31412220). Moreover, 2 separate experimental approaches clearly strengthen the findings of a study.

Moreover, the authors correctly indicate that KP tumors have been shown to be refractory. However, not necessarily in the subcutaneous and orthotopic setting (where the authors find differences on the efficacy of anti-CD40/FLT3L therapy) and in their KP-Neo model (apart from anti-PD-L1 in S.C. tumors Fig. 2A). Why was the old Figure 5 removed from the manuscript that compared anti-PD-L1 and Flt3L/aCD40 in orthotopic KPneo tumors – this Figure just required a more comparable treatment regimen of both therapies. Is it possible that orthotopically (the most relevant setting) grafted KP-neo tumors respond equally to DC-therapy and immune checkpoint blockade when anti-PD-L1 treatment is started earlier and that there may be a synergistic effect of both therapies? This seems highly likely because: (1) DC expansion appears to be equal in KP-control and KP-neo settings upon CD40/Flt3L therapy, but a therapeutic benefit reducing tumour load was only seen in orthotopic KP-neo tumors; (2) Orthotopic KP-neo tumors have a generally enhanced T cell infiltration compared with KP-control tumors (new Fig. 5); (3) the authors state themselves that “DC therapy relieves exhaustion of CD8+ T cells in lung orthotopic tumors”, which is the main mechanism of action of immune checkpoint blockade.

The authors perform most of their analysis on the changes of the DC compartment (Fig. 6 and 7) in orthotopic KP-neo tumours and it is unclear if a similar DC remodelling also occurs in KP-control tumors and is therefore the main driver of the therapeutic effect – or the alterations in the T cell compartment in KP-neo versus KP-control tumors are the critical factor – or both. Hence, conceptually, it is important to know if lung-orthotopic KP-neo

tumors are refractory to immune checkpoint blockade and, hence, DC therapy breaks (immune) therapeutic resistance of lung tumors with a high TMB (which was the main focus of the initial manuscript).

Here my detailed comments on the summary of the authors, which outlines a certain lack of novelty of the revised manuscript:

“In summary, the new results presented in revised Figure 2 and 5 show that:

1) DC therapy given as single therapy strongly inhibits the growth of highly immunogenic tumors in s.c. setting and reduces their progression in orthotopic settings.”

I fully agree. However, the anti-tumor effect of Flt3L + CD40 agonists is known since the 90s (PMID: 10415026). Moreover, since the first submission of this manuscript, the induction of neoantigen expression in KP cells was shown to limit tumor growth in a dose-dependent manner - which was dependent on cDC1s (PMID: 37548358).

“2) Poorly immunogenic tumors respond partially to DC therapy in s.c. setting and are refractory in the immunosuppressive lung environment.”

It was recently shown that anti-CD40+Flt3L is an effective treatment of the mouse model used by the authors: orthotopic KP tumours in the lung independently of TMB (PMID: 34534439).

“3) Recruitment of CD8 T cells in tumors and specific responses to neoAgs are consistent with the therapeutic efficacy and demonstrate that tumor control is mediated by a DC1-CD8 axis, which is more effective in a neoAgs rich context.”

This finding is in contrast with a previous study that claims for a better immunogenicity of a neoAG low, but not high, context in the KP model (PMID: 37548358). Could the authors comment on those differential findings?

“4) An extended analysis of the specificity of CD8 T cell responses, including shared neoAgs, unveiled that: 1) less immunogenic epitopes in KPcont become immunogenic in the KPneo context (Revised Fig1E-G) ; 2) DC1 are required to promote broad responses to both shared and unique neoAgs (Figure 1H); 3) The efficacy of DC-therapy in KPneo depends on DC1!

Please see my responses to the previous points 1-3.

Major comment 2 and 3

It is very much appreciated that the authors convincingly addressed those questions. I may suggest the authors to consider to add the Figure in the P&P to the manuscript? It seems to be relevant to know that the reduction of CCR7+/PD-L1+ DCs is not dependent on neoantigen load of the tumor type.

Major comment 4 – The authors unfortunately solely focus on CD8+ T cells and ignore the emerging role of CD4+ Th1 cells in anti-cancer immunity and the importance of cDC1s controlling their activation and vice versa (PMID: 32457487; 32788723 and 30961656).

Unfortunately, this comment was rather poorly addressed – although some suggestions simply included the re-analysis of already existing data that the authors already generated. This is very surprising due to the relevance of CD4+ T cells and their interaction with cDC1s in cancer (please see my initial comments for more details) and considering that reviewer 2 expressed the same comment.

Reviewer #2 (Remarks to the Author):

I have no further concerns.

Reviewer #3 (Remarks to the Author):

In this revised version of the manuscript, Lopez-Rodriguez et al. have experimentally addressed most of the points raised by the reviewers. In doing so, the study has improved in quality and clarity showing a clear, but not unexpected, role for cDC1s in broadening the anti-tumour T cell response to both unique and shared neoantigens in one transplantable mouse model of lung cancer.

Reviewer #4 (Remarks to the Author):

In the manuscript entitled "DC targeted therapy breaks immune tolerance against neoantigen in refractory lung adenocarcinoma" Lopez-Rodriguez et al used single-cell RNA sequencing to identify the mechanisms of Flt3L therapy in hypermutated cancers. My comments were restricted to the content and technical aspects of the transcriptomic analysis and I evaluated only their responses to these points.

They addressed my comments with an adjusted supplementary figure 4, however there are some types in the gene names in S4C. Please correct these.

The authors did not address one of my points. They mistakenly wrote : "We cannot apply statistics as single-cell data are single points.". This is correct, but that is not what it was requested. Statistical analysis between population abundancies take into consideration the distribution of all cell types.

I had included reference to one of such methods and below I provide additional ones.

Lin, X., Chau, C., Ma, K. et al. DCATS: differential composition analysis for flexible single-cell experimental designs. *Genome Biol* 24, 151 (2023). <https://doi.org/10.1186/s13059-023-02980-3>

Zhao et al 2023 <https://doi.org/10.1073/pnas.2100293118>

The authors also incorrectly did not address an additional point: "We cannot show dot plot as pre and post therapy, as genes are differentially expressed in the various clusters and the intercluster differences would hide intracluster variations. We have now added functional validation of the proliferative phenotype pre and post therapy in revised Figure 6. "

Of course the genes can be displayed and its interpretation should be made clear in the text. It is good they added the phenotype validation, but it does not mean this point cannot be addressed.

Point 5 has also not been properly addressed. I believe there is a misunderstanding on terminology. If they combined expression of multiple genes together using their mean, this will be a score (in this case a linear score with equal weights to all genes). Terminology should be used adequately.

"The values are not scores but mean expression, legend have been modified accordingly: Violin plots show the distribution of log-transformed means of normalized expression values of cytotoxic T cell markers (Gzma, Gzmb, Prf1, Tnf, Ifng, Cxcr6) or exhaustion markers (Pdc1, Ctla4, Tigit, Lag3, Havcr2, Tox, Nrp1) in cells of the indicated clusters in FL/aCD40 vs. control. "

Reviewer's Comments:

Reviewer #1 (Remarks to the Author)

The manuscript Lopez-Rodriguez et al underwent a major revision that importantly changed the focus of the study. This includes notable amounts of new experimental data, but unfortunately also the removal of conceptually most novel/relevant data. Certainly, the authors improved the quality of the designs and related conclusions of selected experimental settings, which is highly appreciated. Unfortunately, the probably most relevant finding of the manuscript (that TMB-high NSCLC tumors are refractory to immune checkpoint blockade, but respond to DC expanding/targeting-therapies that remodel DC populations in the tumor-bearing lung) seems to be even less investigated in the revised compared with the initial manuscript. Instead, the authors focused on the relevance of Flt3L+anti-CD40 therapy for lung cancer with/without high TMB. This change of focus of the manuscript is clearly reflected in the change of the title from “DC targeted therapy breaks immune tolerance against neoantigen in refractory lung adenocarcinoma” to “DCs targeted therapy expands CD8 T cell responses to bona-fide neoantigens in lung tumors”. In light of pre-existing literature (for example PMID: 10415026, 34534439) and a report since first manuscript submission (PMID: 37548358), this new conceptual direction has notably lowered the novelty of the manuscript in my view. I would recommend maintaining the initial message of the manuscript (DC therapy being more effective/enabling immune checkpoint efficacy in otherwise refractory high-neoantigen lung cancers) to reach the relevance required for publication in Nature Communication. This includes the side-by-side comparison of immune checkpoint and DC therapy in lung KP-neo tumors (and ideally the combo therapy) as the authors showed in old Figure 5 (but with comparable aPD-L1 and Flt3L/aCD40 treatment regimens) and was suggested in my Major comment 1 in the course of the first revision.

R: DC-based therapies (including Flt3L ligand and treatments to activate DCs) are actively being explored (these recent reports add to the already cited literature: PMID: 37996514, PMID: 38118422), despite being an old concept. Notably, in the case of lung cancer, DC-based therapies await preclinical validation in relevant antigenic contexts and the underlying molecular mechanisms remain poorly investigated.

In contrast, several papers have already investigated checkpoint inhibition in the most accredited NSCLC model (KP), in various antigenic contexts and with diverse ICB combinations (PMID: 34714687, 34380768, 34738089, 37709863). Collectively these studies provide a strong demonstration that KP are refractory to this class of treatment.

Nevertheless, since our model may be different, we indeed show that it is still refractory to PD-L1 (Figure 2 A, repeated ICB administration from early time points and comparably to DC-therapy), as control. We do not see how further investigations in this respect may have increased the novelty.

Major comment 1 – The responses of NSCLC with or without TMB to immune checkpoint therapy and DC-expanding/targeted therapy is poorly addressed. In my view, the differential response to those immunotherapies and a potential dependence on high-TMB probably represents the most novel and relevant finding of this study and, therefore, should be properly and systematically investigated.

As stated above, our data are clearly showing that both TMB high and low in our model are refractory to ICB (Figure 2A). Other reports using highly immunogenic antigens in KP orthotopic and autochthonous, under diverse ICB regimens reached the same conclusions (PMID: 34714687, 34380768, 34738089, 37709863).

It was very much appreciated that the authors used two independent strategies to deplete cDC1s (Batf3-ko and XCR1-DTA) and enhance DC expansion/efficacy (polyI:C+Flt3L and CD40+Flt3L), despite using only one grafted cancer model in mice. Unfortunately, in the revised manuscript, this was reduced to 1 cancer model, 1 DC-deficiency model and 1 DC expansion strategy. Moreover, those DC-targeted strategies are suboptimal and not specific for cDC1s: the Batf3-ko model also targets CD8+ and regulatory T cells (for example PMID: 29147008, 32989328 and 32669309) and anti-CD40 antibodies as well as Flt3L target/expand several DC subsets and other cells (PMID: 33810248 and 31412220). Moreover, 2 separate experimental approaches clearly strengthen the findings of a study.

Poly:IC initially presented in the study cannot be delivered systemically, this is why we have excluded it from further experiments. Moreover, this activatory ligand has been dropped as several toxic effects have emerged (unpublished data from collaborators to this study). Despite some concerns, Batf3 ko mice are still considered the golden standard for DC depletion (PMID: 38118422, 37548358, 37864794).

As specified in the letter, we could not have access to the XCR1-DTA strain during the revision to test the new treatment regimen. Yet we feel very confident that the results seen in the original experiment with XCR1-DTA very much reflect those obtained with Batf3 null mice.

Moreover, the authors correctly indicate that KP tumors have been shown to be refractory. However, not necessarily in the subcutaneous and orthotopic setting (where the authors find differences on the efficacy of anti-CD40/FLT3L therapy) and in their KP-Neo model (apart from anti-PD-L1 in S.C. tumors Fig. 2A). Why was the old Figure 5 removed from the manuscript that compared anti-PD-L1 and Flt3l/aCD40 in orthotopic KPneo tumors – this Figure just required a more comparable treatment regimen of both therapies. Is it possible that orthotopically (the most relevant setting) grafted KP-neo tumors respond equally to DC-therapy and immune checkpoint blockade when anti-PD-L1 treatment is started earlier and that there may a synergistic effect of both therapies?

We have added back the data showing refractoriness to checkpoint inhibitors in the context of orthotopic tumors, adding the KP^{ctrl} group that was missing in the initial version. Data in Figure S3A show that KP^{ctrl} and KP^{neo} in the orthotopic lungs do not show significant differences in tumor burden upon checkpoint inhibition.

This seems highly likely because: (1) DC expansion appears to be equal in KP-control and KP-neo settings upon CD40/Flt3L therapy, but a therapeutic benefit reducing tumour load was only seen in orthotropic KP-neo tumors; (2) Orthotopic KP-neo tumors have a generally enhanced T cell infiltration compared with KP-control tumors (new Fig. 5); (3) the authors state themselves that “DC therapy relieves exhaustion of CD8+ T cells in lung orthotopic tumors”, which is the main mechanism of action of immune checkpoint blockade.

The authors perform most of their analyse on the changes of the DC compartment (Fig.6 and

7) in orthotopic KP-neo tumours and it is unclear if a similar DC remodelling also occurs in KP-control tumors and is therefore the main driver of for the therapeutic effect – or the alterations in the T cell compartment in KP-neo versus KP-control tumors are the critical factor – or both. Hence, conceptually, it is important to know if lung-orthotopic KP-neo tumors are refractory to immune checkpoint blockade and, hence, DC therapy breaks (immune) therapeutic resistance of lung tumors with a high TMB (which was the main focus of the initial manuscript).

We have added data showing that DC-therapy expands DCs regardless of the tumor burden (Figure 5C and S6E). This is an expected result as Flt3L acts on the bone marrow to promote pre-DC output, independently of CD8 T cells. Therefore our model proposes that DC are expanded equally in neoAgs low and high tumors. However, only in the presence of sufficient neoAgs encoded in cancer cells, expanded DC expand and activate the corresponding neoAgs specific T cells. In neoAgs low settings, the lack of sufficient neoAgs remains the limiting factor and DC expansion cannot compensate for it. Strengthening this view, Flt3L has no impact on T cells when DC1 are missing. We have discussed this concept at page 20 and 26 in the revised text.

Here my detailed comments on the summary of the authors, which outlines a certain lack of novelty of the revised manuscript:

“In summary, the new results presented in revised Figure 2 and 5 show that:

1) DC therapy given as single therapy strongly inhibits the growth of highly immunogenic tumors in s.c. setting and reduces their progression in orthotopic settings.”

I fully agree. However, the anti-tumor effect of Flt3L + CD40 agonists is known since the 90s (PMID: 10415026). Moreover, since the first submission of this manuscript, the induction of neoantigen expression in KP cells was shown to limit tumor growth in a dose-dependent manner - which was dependent on cDC1s (PMID: 37548358).

The elegant paper by Spranger group hereby cited investigates the role of high or low intratumoral heterogeneity, inserting 2 model antigens together or separately in KP clones and does not address the role of DC therapy on the response. Our study instead analyses responses to weak and strong bona fide neoAgs encoded clonally by the same cancer cell and explores how DC therapy/DC depletion modifies the pattern of responding antigens.

“2) Poorly immunogenic tumors respond partially to DC therapy in s.c. setting and are refractory in the immunosuppressive lung environment.”

It was recently shown that anti-CD40+Flt3L is an effective treatment of the mouse model used by the authors: orthotopic KP tumours in the lung independently of TMB (PMID: 34534439).

The paper by T.Jacks group, has been cited repeatedly in the paper to indicate that indeed major groups are moving to DC-therapy away from ICB for lung cancer. This study uses the single model antigen OVA and does not enter into the complexity of responses to multiple neoAgs nor the mechanism underlying DC therapy.

“3) Recruitment of CD8 T cells in tumors and specific responses to neoAgs are consistent with the therapeutic efficacy and demonstrate that tumor control is mediated by a DC1-CD8 axis, which is more effective in a neoAgs rich context.”

This finding is in contrast with a previous study that claims for a better immunogenicity of a neoAG low, but not high, context in the KP model (PMID: 37548358). Could the authors comment on those differential findings?

The study cited does not show that low immunogenicity is better in KP. Instead, the authors engineer two model antigens (one weak and one strong), expressed together or separately in cancer lines. The results show that the weak antigen induces responses only when co-expressed with the strong antigen.

This is very much in line with our data showing that shared neoAgs do not induce responses in KP^{ctrl} but they do in KP^{neo}, expanding the context to bona-fide neoAgs

“4)An extended analysis of the specificity of CD8 T cell responses, including shared neoAgs, unveiled that: 1) less immunogenic epitopes in KP^{cont} become immunogenic in the KP^{neo} context (Revised Fig1E-G) ; 2) DC1 are required to promote broad responses to both shared and unique neoAgs (Figure 1H); 3) The efficacy of DC-therapy in KP^{neo} depends on DC1! Please see my responses to the previous points 1-3.

Major comment 2 and 3

It is very much appreciated that the authors convincingly addressed those questions. I may suggest the authors to consider to add the Figure in the PxP to the manuscript? It seems to be relevant to know that the reduction of CCR7+/PD-L1+ DCs is not dependent on neoantigen load of the tumor type.

R: Thank you for the suggestion. We have added the data in Figure S6E and discussed that DC driven remodeling is antigen-agnostic in the text at page 20 and 26.

Major comment 4 – The authors unfortunately solely focus on CD8+ T cells and ignore the emerging role of CD4+ Th1 cells in anti-cancer immunity and the importance of cDC1s controlling their activation and vice versa (PMID: 32457487; 32788723 and 30961656).

Unfortunately, this comment was rather poorly addressed – although some suggestions simply included the re-analysis of already existing data that the authors already generated. This is very surprising due to the relevance of CD4+ T cells and their interaction with cDC1s in cancer (please see my initial comments for more details) and considering that reviewer 2 expressed the same comment.

Reviewer #2 (Remarks to the Author):

I have no further concerns.

Reviewer #3 (Remarks to the Author)

In this revised version of the manuscript, Lopez-Rodriguez et al. have experimentally

addressed most of the points raised by the reviewers. In doing so, the study has improved in quality and clarity showing a clear, but not unexpected, role for cDC1s in broadening the anti-tumour T cell response to both unique and shared neoantigens in one transplantable mouse model of lung cancer.

Reviewer #4 (Remarks to the Author):

In the manuscript entitled “DC targeted therapy breaks immune tolerance against neoantigen in refractory lung adenocarcinoma” Lopez-Rodriguez et al used single-cell RNA sequencing to identify the mechanisms of Flt3L therapy in hypermutated cancers. My comments were restricted to the content and technical aspects of the transcriptomic analysis and I evaluated only their responses to these points.

They addressed my comments with an adjusted supplementary figure 4, however there are some typos in the gene names in S4C. Please correct these.

R: This has been corrected, thank you for pointing out

The authors did not address one of my points. They mistakenly wrote : "We cannot apply statistics as single-cell data are single points.". This is correct, but that is not what it was requested. Statistical analysis between population abundancies take into consideration the distribution of all cell types.

I had included reference to one of such methods and below I provide additional ones.

Lin, X., Chau, C., Ma, K. et al. DCATS: differential composition analysis for flexible single-cell experimental designs. *Genome Biol* 24, 151 (2023). <https://doi.org/10.1186/s13059-023-02980-3>

Zhao et al 2023 <https://doi.org/10.1073/pnas.2100293118>

As suggested by the Reviewer, we have performed differential abundance analysis using the DA-seq algorithm, which could fit with our experimental design, as both DCATS and miloR require more than one replicate per condition to run the method. We run the DA-seq algorithm on our whole dataset, containing all immune cell types, and we quantified a local measure of differential abundance for each cell based on k nearest neighbors across a range of k values (from 50 to 500, as recommended). We have now added the UMAP embeddings for T cells and DCs (Revised Figure S4A and S6A), colored according to their differential abundance score. This analysis confirms the increased abundance of CL6 of CD8+ T cells after therapy, while no evident differential abundance is observed for CL5 of CD8+ T cells. In the DCs compartment, the new analysis confirms depletion of mregDCs which are more abundant before therapy.

The authors also incorrectly did not address an additional point: "We cannot show dot plot as pre and post therapy, as genes are differentially expressed in the various clusters and the intercluster differences would hide intracluster variations. We have now added functional validation of the proliferative phenotype pre and post therapy in revised Figure 6. "

Of course the genes can be displayed and its interpretation should be made clear in the text. It is good they added the phenotype validation, but it does not mean this point cannot be addressed.

We apologise for the lack of clarity in our reply. We have separated the dot plot in Figure 6A according to treatment groups, as per your suggestion and we provide the data below for the reviewer appraisal. Although the dot sizes indicate an increased expression of proliferation genes upon treatment in the three non-naive CD8+ T cell clusters (C3, C5, C6), the elevated expression of these genes by C6 may underestimate the relevant differential expression between the samples. Given that this panel is intended to display the expression of the most representative genes (marker genes) to identify each cluster, we believe that presenting separate dot plots might be misleading and we prefer to keep the current representation. Comparative analysis of gene expression according to treatment in the different clusters is addressed in the panels that follow.

Point 5 has also not been properly addressed. I believe there is a misunderstanding on terminology. If they combined expression of multiple genes together using their mean, this will be a score (in this case a linear score with equal weights to all genes). Terminology should be used adequately.

We do agree with the reviewer for a possible misunderstanding of the terminology. In the violin plots, we report the mean of expression values of the indicated gene. Specifically, we subset the normalized gene expression matrix on the selected genes and compute the mean expression of the signature for each cell using the colMeans() function. We have now modified the axis (score) and the legend and added a sentence in methods.

Legend: the score represents the mean of expression values of the selected gene signatures.

REVIEWERS' COMMENTS

Reviewer #4 (Remarks to the Author):

I am satisfied with the changes performed by the authors and they successfully addressed all my points.